# On the Regularity of Attention

## Abstract

Attention is a powerful component of modern neural networks across a wide variety of domains. In this paper, we seek to quantify the regularity (i.e. the smoothness) of the attention operation. To accomplish this goal, we propose a new mathematical framework that uses measure theory and integral operators to model attention. Specifically, we formulate attention as an operator acting on empirical measures over representations of tokens. We show that this framework is consistent with the usual definition, captures the essential properties of attention, and that it can handle inputs of arbitrary length. Then we use it to prove that, on compact domains, the attention operation is Lipschitz continuous with respect to the 1-Wasserstein distance, and provide an estimate of its Lipschitz constant. Additionally, by focusing on a specific type of attention, we extend these Lipschitz continuity results to non-compact domains. Finally, we discuss the effects regularity can have on NLP models, as well as applications to invertible and infinitely-deep networks.

## 1 Introduction

Attention (Bahdanau et al., 2014; Vaswani et al., 2017) is a fundamental building block of modern neural networks. However, despite its ubiquity, much is still not well understood about the mathematical properties of attention; in this paper, we study the question of regularity. Regularity (i.e. smoothness) is a fundamental property of neural networks with important implications for topics such as robustness (Virmaux & Scaman, 2018; Finlay et al., 2018; Salman et al., 2019; Bubeck & Sellke, 2021), generalization guarantees (Bartlett et al., 2017; Neyshabur et al., 2018; Chuang et al., 2021), and uncertainty estimation (Liu et al., 2020; van Amersfoort et al., 2021). However, because of some special properties of attention such as self-interaction and the ability to process variable length inputs, special care must be taken to model attention *and* obtain a robust theory. For example, it is not clear *a priori* how to measure the closeness of two inputs to attention that have different numbers of vectors.

In this paper, we address the technical challenges of studying the regularity of attention by formulating attention in terms of measure theory and integral operators. More precisely, we propose a framework where attention is an operator acting on empirical measures over representations of tokens. These measures can encode an arbitrary number of tokens. By equipping this space of measures with the 1-Wasserstein distance, our framework allows us to evaluate the regularity of attention in terms of its Lipschitz continuity (see Figure 1 for an illustration). We investigate the implications of this regularity on a number of concrete scenarios including cross-attention, which has become important in recent model architectures (Jaegle et al., 2021; Alayrac et al., 2022). We also study how it can help certain applications by providing robustness to the learned representations, but hurt others when the regularity of the model does not match the regularity of the task. Lastly, we study how regularity impacts the properties of self-attention networks such as their invertibility and the existence of infinite-depth limits.

The paper is organized as follows: we survey some related work in Section 2, then introduce preliminaries in Section 3 and describe attention using measure theory in Section 4. We then obtain quantitative Lipschitz continuity estimates for self-attention in Section 5 and apply these results to some concrete problems in Section 6. Finally, we and conclude in Section 7.

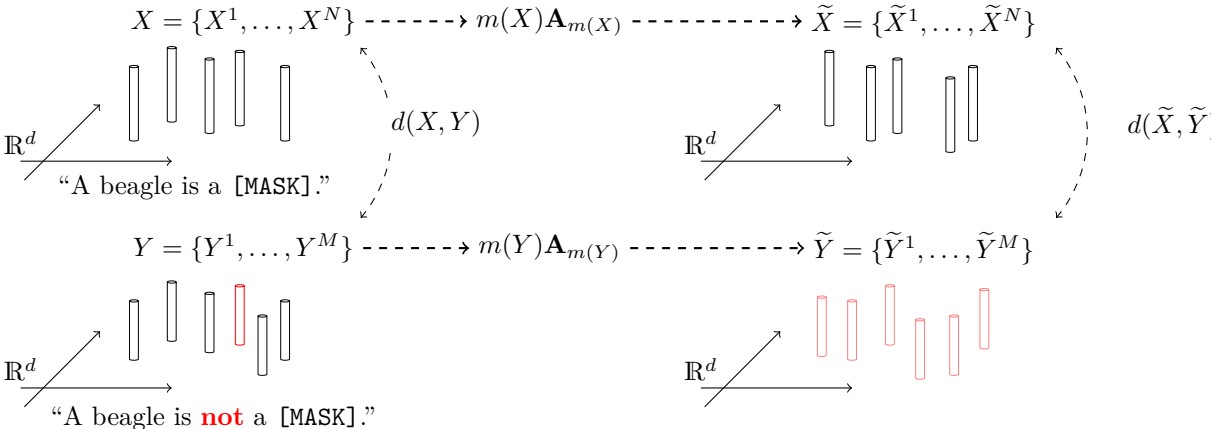

**Figure 1:** Illustration of our framework. A sentence, viewed as a set of vectors in $\mathbb{R}^d$, is represented by its empirical measure on $\mathbb{R}^d$. Above, we visualize two sentences: "A beagle is a `[MASK]`." ($N = 5$) and "A beagle is *not* a `[MASK]`." ($M = 6$). We visualize them by placing a cylinder at the position of the corresponding token's representation in the sentence (the cylinders have a height of either $\frac{1}{N}$ or $\frac{1}{M}$). Self-attention, which transforms $X$ into $\tilde{X}$, is modelled as an operator on the corresponding empirical measures. We equip the space of measures on $\mathbb{R}^d$ with the 1-Wasserstein distance and show that the self-attention operator is Lipschitz: $d(\tilde{X}, \tilde{Y}) \leq \tau(\mathbf{A})d(X, Y)$. In our example, adding *not* to the sentence modifies the output set. However, the extent of the modification is constrained by $\tau(\mathbf{A})$, see additional details in Section 6.

## 2 Related Work

As noted by Smola & Zhang (2019), the original notion of attention appears in statistics in the form of the Watson-Nadaraya estimator (Watson, 1964; Nadaraya, 1964) which implements a data-dependent regression model. The term "attention" and the modern "query-key-value" formulation comes from Bahdanau et al. (2014) who use attention for sequence alignment in a recurrent neural translation model. A similar setup was used in Graves et al. (2014) for differentiable, content-based addressing of a memory array. In Sukhbaatar et al. (2015) and Seo et al. (2016), attention is used for question answering, machine reading comprehension, and language modelling. The extremely successful "Transformer" architecture was introduced in Vaswani et al. (2017) and demonstrated that one could build powerful neural networks using attention as the main component. This led to important developments in language modelling (Devlin et al., 2018; Radford et al., 2018; Raffel et al., 2019; Brown et al., 2020), graph modelling (Veličković et al., 2017), image modelling (Parmar et al., 2018; Dosovitskiy et al., 2020), set modelling (Lee et al., 2018), reinforcement learning (Baker et al., 2019), and multimodal learning Alayrac et al. (2022) among others.

There has also been recent work studying of the properties of attention-based networks from a theoretical perspective. In Kim et al. (2021), the authors study the Lipschitz constant of self-attention as a map from $\mathbb{R}^{d \times N} \to \mathbb{R}^{d \times N}$; we will revisit their approach in Section 6. Dasoulas et al. (2021) propose an explicit normalization scheme for self-attention layers so as to enforce Lipschitz continuity. Other works studying various theoretical aspects of attention (not necessarily regularity) include Edelman et al. (2022); Bhattamishra et al. (2020); Hron et al. (2020); Levine et al. (2020). Another recent work from Dong et al. (2021) shows that the other structural components of transformers (i.e., the feed-forward networks and residual connections) are in fact *necessary* to avoid rank collapse of the resulting representations. This is an interesting finding because our focus is on the attention operation itself, and hence our analysis doesn't include residual connections or feed-forward networks.

We were mathematically inspired by Del Moral (2004) who studied self-interacting "Feynman-Kac models" using semigroup techniques (including contractions for nonlinear operators on measures). An interacting particle interpretation of attention is studied in Lu et al. (2019) using tools from dynamical systems theory.

## 3 Preliminaries

### 3.1 Attention

The fundamental definition of attention is due to Bahdanau et al. (2014), which we provide below with some additional terminology for the various components that we will study.

**Definition 1** (Attention, Bahdanau et al. (2014)). *Let $K = (k_1, \ldots, k_N) \subset \mathbb{R}^{d_k}$ be a collection of **keys**, $V = (v_1, \ldots, v_N) \subset \mathbb{R}^{d_v}$ a collection of corresponding **values**, and $q \in \mathbb{R}^{d_q}$ a **query**. Also, let $a : \mathbb{R}^{d_q} \times \mathbb{R}^{d_k} \to \mathbb{R}$ be a measurable **similarity function**. Then **attention** is the mapping*

$$\text{Attention}(q, K, V) := \sum_{i=1}^{N} \text{softmatch}_a(q, K)_i \cdot v_i,$$

*where $\text{softmatch}_a(q, K)$ is a probability distribution over the elements of $K$ defined as*

$$\text{softmatch}_a(q, K)_i := \frac{\exp(a(q, k_i))}{\sum_{j=1}^{N} \exp(a(q, k_j))}. \tag{1}$$

While $\text{Attention}(\bullet, K, V)$ is defined point-wise for a given query, it is almost always used to process a set of queries $Q = \{q_1, \ldots, q_M\} \subset \mathbb{R}^{d_q}$ in parallel. Thus, we will usually write $\text{Attention}(Q, K, V) := \{\text{Attention}(q_i, K, V)\}_{i=1}^{M}$. Also, while $|K| = |V| = N$, in general $M$ does not have to equal $N$. When $K = V = Q$, we call the following mapping **self-attention**:

$$Q \mapsto \text{SelfAttention}(Q) := \text{Attention}(Q, Q, Q).$$

We are primarily interested in self-attention as it *can be composed to arbitrary depth*, making it a key building block of many neural network architectures.

### 3.2 Markov Kernels

In the sequel, $(E, \mathcal{E})$ denotes a subset of $\mathbb{R}^d$ endowed with its Borel $\sigma$-algebra, and $\mathcal{P}(E)$ the space of probability measures on $E$. We use the following notation for expectations w.r.t. $\mu \in \mathcal{P}(E)$: for a real-valued measurable function $f$, we denote $\mu(f) := \int f(x)\mu(\mathrm{d}x)$ when it exists.

Our framework will heavily rely on linear transformations of measures modelled by Markov kernels; see e.g. Del Moral (2004) for an account that is consistent with our notation.

**Definition 2** (Markov kernel). *A **Markov kernel** is a mapping $M : E \times \mathcal{E} \to [0, 1]$ such that $\forall x \in E, M(x, \bullet) \in \mathcal{P}(E)$ and $\forall A \in \mathcal{E}, x \mapsto M(x, A)$ is measurable.*

A Markov kernel $M$ defines a linear operator $\mathcal{P}(E) \to \mathcal{P}(E)$ by $\mu M(\mathrm{d}y) := \int \mu(\mathrm{d}x) M(x, \mathrm{d}y)$. It also defines a linear operator on measurable functions by $M(f)(x) := \int f(y) M(x, \mathrm{d}y)$. Markov kernels $M, N$ can be composed by integration, $MN(x, \mathrm{d}z) := \int M(x, \mathrm{d}y) N(y, \mathrm{d}z)$.

## 4 Modelling Attention

In this section, we model attention (Bahdanau et al., 2014) and the Transformer (Vaswani et al., 2017) in measure-theoretic language. Our construction casts the action of attention on collection of vectors as a nonlinear Markov transport on $\mathcal{P}(E)$ by reformulating existing linear algebra and point-wise operations in-terms of operators on $\mathcal{P}(E)$.

### 4.1 Basic Model of Attention

The fundamental parts of Attention from Definition 1 are: the $\text{softmatch}_a$ operation, the key-value correspondence, and the value-averaging w.r.t. the softmatch distribution. We will treat each of these in turn.

**Softmatch and Boltzmann-Gibbs Transformations.** At the core of the softmatch function, and indeed attention itself, are the interactions between queries and keys. These interactions are a specific case of a nonlinear measure transformation, the Boltzman-Gibbs transformation.

**Definition 3** (Boltzmann-Gibbs Transformation). *Let $g : E \to \mathbb{R}_{>0}$ be bounded and measurable. The* **Boltzmann-Gibbs transformation** *associated to $g$ is the mapping $\Psi_g : \mathcal{P}(E) \to \mathcal{P}(E)$:*

$$\Psi_g(\nu)(dy) := \frac{g(y)\nu(dy)}{\nu(g)}.$$

To implement the $\text{softmatch}_a$ operation, we will need a function $G : E \times E \to \mathbb{R}_+^*$ taking the form $G(x, y) = \exp(a(x, y))$, where $a$ is a similarity function as in Definition 1. We call $G$ an **interaction potential**.

**Definition 4** (Softmatch Kernel). *For an interaction potential $G$, we call the* **softmatch kernel** *the family of Markov kernels $\{\Psi_G(\nu)\}_{\nu \in \mathcal{P}(E)}$ indexed by $\nu \in \mathcal{P}(E)$, such that for $A \in \mathcal{E}$*

$$\Psi_G(\nu)(x, A) = \int_A \Psi_{G(x,\bullet)}(\nu)(dy) = \frac{\int_A G(x, y)\nu(dy)}{\int_E G(x, y)\nu(dy)}.$$

In other words, for a given $x \in E$ and $\nu \in \mathcal{P}(E)$, the softmatch kernel $\Psi_G(\nu)(x, dy)$ is the Boltzmann-Gibbs transformation associated to $G(x, \bullet)$. To see how $\Psi_G$ can be used to model the softmatch operation, we introduce some simple but useful constructions from measure theory.

**Empirical measure mapping.** Denote by $\mathcal{P}_\delta(E) := \{\delta_x \mid x \in E\}$ the subset of **Dirac measures** in $\mathcal{P}(E)$. There is a natural bijection between $E$ and $\mathcal{P}_\delta(E)$ defined by $x \leftrightarrow \delta_x$ which will be the primary entry point for measure theory in our model of attention. We can associate to any set of vectors $X = \{x_1, \ldots, x_N\} \subseteq E \subseteq \mathbb{R}^d$ a measure in $\mathcal{P}(E)$ via the **empirical measure mapping**:

$$X \mapsto m(X) := \frac{1}{N} \sum_{i=1}^{N} \delta_{x_i}.$$

In what follows, we will often use $X$ and $\{\delta_{x_1}, \ldots, \delta_{x_N}\}$ interchangeably to represent the individual vectors and $m(X)$ to represent the joint configuration of $X$. We will see below that $m(X)$ is a very natural object to represent this joint configuration and how it behaves with attention.

Now consider a "query" representation $\delta_q$, "key" representations $K = \{\delta_{k_1}, \ldots, \delta_{k_N}\}$, and the empirical measure $m(K)$. The softmatch kernel models the interaction between $q$ and $K$ using the left-action of the Markov kernels $\Psi_G(m(K))$ on the Dirac measure $\delta_q$ induced by integration:

$$\delta_q \Psi_G(m(K)) = \int \delta_q(dq') \Psi_{G(q',\bullet)}(m(K)) = \sum_{s=1}^{N} \frac{G(q, k_s)}{\sum_{r=1}^{N} G(q, k_r)} \delta_{k_s}.$$

Furthermore, given a set of queries $Q = \{\delta_{q_1}, \ldots, \delta_{q_M}\}$, we can leverage the linearity of integration to model the interaction between the two sets of representations $Q$ and $K$ using the same principle:

$$m(Q) \Psi_G(m(K)) = \frac{1}{M} \sum_{t=1}^{M} \int \delta_{q_t}(dq) \Psi_{G(q,\bullet)}(m(K)) = \frac{1}{M} \sum_{t=1}^{M} \sum_{s=1}^{N} \frac{G(q_t, k_s)}{\sum_{r=1}^{N} G(q_t, k_r)} \delta_{k_s}.$$

This new measure represents the joint configuration of the set of queries $Q$ *after* they have interacted with the keys $K$ through the potential $G$ and the associated Boltzmann-Gibbs transformation. It is a *weighted sum* of particle measures, and will allow us to model the softmatch operation from Eq. (1).

**Key-Value Relationships.** To generalize the relationship between keys and values, we now introduce the lookup kernel.

**Definition 5** (Lookup Kernel). *Assume that the keys and values come from (Borel) measurable subsets of $\mathbb{R}^{d_k}, \mathbb{R}^{d_v}$ resp. A lookup kernel is a Markov kernel, $L : \mathbb{R}^{d_k} \times \mathcal{B}(\mathbb{R}^{d_v}) \to [0,1]$, also denoted $L(k, \mathrm{d}v)$, that maps keys to distributions on values. When the mapping from keys to values is a deterministic function $\ell$, we have $L(k, \mathrm{d}v) = \delta_{\ell(k)}(\mathrm{d}v)$.*

For self-attention, $\ell(x) = x$ is the natural choice of the deterministic lookup function, and for the Transformer (see App. A), the natural choice is $\ell(k) = W^V k$. In general, to study regularity, we assume there exists *some* well-behaved function $\ell : \mathbb{R}^{d_k} \to \mathbb{R}^{d_v}$ that realizes the correspondence $k_i \leftrightarrow v_i$ — this holds for most realistic implementations of attention such as those above.

**Remark 1.** *The most general case of attention, when there is no prescribed correspondence between $k_i$ and $v_i$, could be realized by a function such as*

$$\ell(k) = \sum_{i=1}^{N} \mathbb{1}_{\{k=k_i\}} v_i.$$

*but this is not in general regular without additional assumptions.*

**Averaging and Measure Projections.** In the remainder of this paper, we will make the following technical assumption, which ensures that the operations we describe are well-defined.

**Assumption 1.** $E \subset \mathbb{R}^d$ *is convex.*

The final element of our construction is the averaging w.r.t. the set of values.

Denote by $\Pi : \mathcal{P}(E) \to \mathcal{P}_\delta(E)$ the **measure projection** of a probability measure $\mu \in \mathcal{P}(E)$ onto the subset of Dirac measures $\mathcal{P}_\delta(E)$ defined by

$$\Pi[\mu] := \delta_{\overline{\mu}}, \quad \overline{\mu} := \int x\mu(\mathrm{d}x) \in E \tag{2}$$

whenever $\overline{\mu}$ exists (e.g. when $\mu$ has finite first moments). We claim (to be justified in a moment) that the averaging w.r.t. values is accomplished by the measure projection $\Pi$ described in Eq. (2).

**The Attention Kernel.** Combining these, we obtain a model for attention, the attention kernel.

**Definition 6** (Attention Kernel). *The **attention kernel**, denoted $\mathbf{A}$, is the composition of the measure projection $\Pi$, the softmatch kernel and the lookup kernel, defined for $q \in E$ and $\mu \in \mathcal{P}(E)$ as:*

$$\mathbf{A}_\mu(q, dz) := \Pi[\Psi_{G(q, \bullet)}(\mu)L](dz) = \Pi\left[\int \Psi_{G(q, \bullet)}(\mu)(\mathrm{d}k)L(k, \mathrm{d}v)\right](\mathrm{d}z),$$

where the softmatch and lookup kernels are composed by integration as described after Definition 2 and $\Pi$ is applied to the resulting measure (which is defined per $q$). Our first result is that this attention kernel is consistent with attention from Definition 1, for suitable choices of $G$ and $L$.

**Proposition 1.** *Let $G(x, y) = \exp(a(x, y))$, $L(k, \mathrm{d}v) = \delta_{\ell(k)}(\mathrm{d}v)$, and $Q, K, V$ be as in the definition of attention. Then, using the left action of kernels on measures, the mapping:*

$$(Q, K, V) \mapsto \left\{\delta_{q_1}\mathbf{A}_{m(K)}, \ldots, \delta_{q_T}\mathbf{A}_{m(K)}\right\}$$

*implements attention as in Definition 1.*

*Proof.* Using the remarks from earlier, for $q \in \mathbb{R}^{d_q}$, we have:

$$\Psi_{G(q, \bullet)}(m(K))L = \int \sum_{j=1}^{N} \frac{G(q, k_j)}{\sum_{p=1}^{N} G(q, k_p)} \delta_{k_j}(\mathrm{d}k)L(k, \mathrm{d}v) = \sum_{j=1}^{N} \frac{G(q, k_j)}{\sum_{p=1}^{N} G(q, k_p)} \delta_{v_j}(\mathrm{d}v).$$

Applying $\Pi$ yields: $\mathbf{A}_{m(K)}(q, \mathrm{d}v) = \delta_{\sum_{j=1}^{N} \frac{G(q,k_j)}{\sum_{p=1}^{N} G(q,k_p)} v_j}(\mathrm{d}v)$. Using the (linear) left-action of this kernel on $\delta_{q_t}$, we then obtain:

$$\delta_{q_t} \mathbf{A}_{m(K)}(\mathrm{d}v) = \int \delta_{q_t}(\mathrm{d}q) \mathbf{A}_{m(K)}(q, \mathrm{d}v) = \delta_{\sum_{j=1}^{N} \frac{G(q_t,k_j)}{\sum_{p=1}^{N} G(q_t,k_p)} v_j}(\mathrm{d}v).$$

Plugging in the definition of $G$ and using the usual bijection $\delta_x \leftrightarrow x$ concludes the proof. $\square$

**Attention as a System of Interacting Particles.** Let us step back and understand the attention kernel $\mathbf{A}$ from a higher level. Consider self-attention: we have effectively factorized the original, linear-algebraic self-attention operation into a series of measure transformations:

$$E \xrightarrow{x \mapsto \delta_x} \mathcal{P}_\delta(E) \xrightarrow{\Psi_G L} \mathcal{P}(E) \xrightarrow{\Pi} \mathcal{P}_\delta(E) \xrightarrow{\delta_x \mapsto x} E.$$

More importantly, we *have a closed-form expression for the evolution of the joint configuration $m(Q)$ of $Q$,* i.e. $m(Q) \mapsto m(Q)\mathbf{A}_{m(Q)}$. Since interaction with the joint configuration is central to attention, having a framework that describes its evolution will be vital to further analysis.

Moreover, as we noted earlier, self-attention can be composed arbitrarily. Indeed, let $Q^0 := Q$ and consider the evolution of a the set of "particles" $Q^h = \{\delta_{q_1^h}, \ldots, \delta_{q_M^h}\}$ for $h = 0, 1, 2, \ldots, H-1$ whose dynamics are given by

$$q_i^{h+1} \sim \mathbf{A}_{m(Q^h)}^h(q_i^h, \bullet)$$

or equivalently as a measure-valued equation

$$\delta_{q_i^{h+1}} = \delta_{q_i^h} \mathbf{A}_{m(Q^h)}^h.$$

Our framework shows that self-attention networks are actually simulating deterministic interacting particle systems for a finite number of time steps corresponding to the number of layers $H$. The representations one obtains are the states of the system after $H$ steps of the dynamics.

**Remark 2.** *Interestingly, the particle interpretation above is studied in Lu et al. (2019) using tools from dynamical systems theory. The authors recognize the Transformer (with the residual connection) as a coupled system of particles evolving under diffusion-convection ODE dynamics, and study this system using the a numerical scheme for the underlying ODE.*

**Remark 3** (Connection with Expectation)**.** *Let us also point out a connection with Bayesian statistics: when $G(q, \bullet) = p(q|\bullet)$ is a likelihood function, $\nu \mapsto \Psi_{G(q,\bullet)}(\nu)$ is the mapping which takes a prior distribution $\nu(\mathrm{d}k)$ over keys and returns a posterior distribution $P(\mathrm{d}k|q)$. Moreover, assuming that $q \to k \to v$ forms a Markov chain, $\Psi_G L(q, \mathrm{d}v)$ models the conditional probability of $v|q$. Finally, the measure projection operator effectively reduces this to a measure concentrated on a single point, $\mathbb{E}[v|q]$, which is consistent with the existing interpretation of attention.*

**Recovering the Traditional Definition.** We have introduced a framework for attention-based models that uses measure theory and Markov kernels as the principal building blocks, and we have shown that it is equivalent (i.e. Proposition 1). It is reasonable to wonder if there is a way to recover the traditional linear-algebraic definition of attention from our framework, and the answer is yes.

Before proceeding, recall three basic facts about Markov kernels and measures on *discrete* spaces:

1. probability measures are stochastic vectors,
2. Markov kernels are stochastic matrices, and
3. integration against these kernels is matrix multiplication.

Suppose now that $E$ is the discrete set $E = \{1, \ldots, N\}$ and let $Q = \{q_1, \ldots, q_N\}$, $K = \{k_1, \ldots, k_N\}$, and $V = \{v_1, \ldots, v_N\}$ be subsets of $\mathbb{R}^d$. Then we set $G : Q \times V \to \mathbb{R}_{\geq 0}$ to be $G(i,j) := \exp \langle Q[i], K[j] \rangle$, $L : K \times 2^V$ is the (discrete) Markov kernel $L(i, \mathrm{d}j) = \delta_i(\mathrm{d}j)$ and $\Pi : \mathcal{P}(V) \to \mathbb{R}^d$ is defined by $\Pi(\mu) := \sum_{j=1}^N V[j]\mu_j =: V'[i]$. Then the analog of the attention kernel is the attention *map*

$$\mathbf{A}_{m(K)}(i) := \Pi[\Psi_{G(i,\bullet)}(m(K))L].$$

Having now defined discrete analogs of the components of attention defined in this section, we can make the following remarks. Firstly, $\delta_{q_i}(\mathrm{d}j)$ is the Kronecker delta $\delta_i^j$ and $m(K) = \frac{1}{N}\mathbf{1}$ where $\mathbf{1} := [1, \ldots, 1]$ (i.e. $N$ times). Secondly, the Boltzmann-Gibbs Markov kernel $\Psi_{G(i,\bullet)}(\mathrm{d}j)$ is then equal to the usual softmax definition:

$$\Psi_{G(i,\bullet)}(m(K))(\mathrm{d}j) = \mathrm{softmax}(Q[i]K^T).$$

The softmatch kernel is the stochastic matrix $\Psi_G(m(K)) = \mathrm{softmax}(QK^T)$. Thirdly, composition of kernels is matrix multiplication, so $\Psi_G(m(K))L = \mathrm{softmax}(QK^T)Id = \mathrm{softmax}(QK^T)$ since $L$ is the $N \times N$ identity matrix $Id$. Finally, the attention map corresponds to the matrix multiplications

$$\delta_{Q[i]}\mathbf{A}_{m(K)} = \Pi[\Psi_{G(i,\bullet)}(m(K))L]$$
$$= \sum_{j=1}^n \Psi_{G(i,\bullet)}(m(K))(\mathrm{d}j)V[j] = \sum_{j=1}^n \mathrm{softmax}(Q[i]K^T)[j]V[j] = \mathrm{softmax}(Q[i]K^T)V.$$

This is the attention definition from Bahdanau et al. (2014) and is used widely in the machine learning community.

We feel that using measures on the representation space rather than the index space offers significant gains. The reason for this is that distributions on $\mathbb{R}^d$ are much more expressive than distributions on $\{1, \ldots, N\}$, and therefore admit more interesting analysis; see for example our analysis of the regularity of attention in Section 5 relies on the 1-Wasserstein distance, which is trivial in the discrete case.

### 4.2 Extension to the Transformer

We now sketch how to extend the measure-theoretic model of self-attention described in the previous section to the popular Transformer encoder architecture (Vaswani et al., 2017). It is a straightforward application of the techniques above. We only describe here how our framework can model a single head Transformer[1], and refer the interested reader to Appendix A for the extension to a full multi-headed Transformer. We seek to model

$$\mathrm{Transformer}(X) = \mathrm{FFN} \circ \mathrm{SelfAttention}(X), \tag{3}$$

where $X = \{x_1, \ldots, x_N\} \subset \mathbb{R}^d$ is the input data, $\mathrm{SelfAttention}(\bullet)$ is the scaled dot-product attention (Vaswani et al., 2017) and $\mathrm{FFN}(\bullet)$ represents a feedforward neural network. We set $\widetilde{a}(x, y) = x^T y / \sqrt{d}$ and let

$$a(x,y) = \widetilde{a}\left(W^Q x, W^K y\right), \qquad L(k, \mathrm{d}v) = \delta_{W^V k}(\mathrm{d}v),$$

where $W^Q, W^K, W^V$ are matrices in $\mathbb{R}^{d \times d}$. These correspond to the various matrix operations performed by the Transformer. We let $f : E \to E$ be the FFN in (3) and define the FFN kernel as $\mathbf{F}(x, \mathrm{d}y) = \delta_{f(x)}(\mathrm{d}y)$. Using the attention kernel $\mathbf{A}$ from Definition 6, we define $\mathbf{T} := \mathbf{AF}$, and show in the proposition below that $\mathbf{T}$ implements the self-attention transformer (proof in Appendix A).

**Proposition 2.** *Let $X = \{x_1, \ldots, x_N\} \subset \mathbb{R}^d$ be a collection of inputs. The nonlinear Markov transport equation $\delta_{x_i} \mapsto \delta_{x_i} \mathbf{T}_{m(X)}$ implements the self-attention Transformer.*

## 5 Regularity of Attention

In this section, we consider self-attention as a non-linear map from $\mathcal{P}(E)$ to $\mathcal{P}(E)$ through $\mathbf{A} : \mu \to \mu\mathbf{A}_\mu$. To derive a Lipschitz contraction estimate, we must first metrize $\mathcal{P}(E)$.

---

[1]We only consider the encoder part of the transformer, since it uses self-attention. Our framework is fully compatible with the cross-attention from the transformer decoder (Vaswani et al., 2017), see Section 6.1.

**Background.** We will work with the Wasserstein metric on $\mathcal{P}(E)$. Let $\mathcal{P}_1(E)$ be the set of probability measures with finite 1st moment. The **1-Wasserstein distance** between $\mu, \nu \in \mathcal{P}_1(E)$ is

$$\mathrm{W}_1(\mu, \nu) := \sup_{f \in Lip_1(E)} \left| \int f \mathrm{d}\mu - \int f \mathrm{d}\nu \right|.$$

$\mathrm{W}_1$ is a metric on $\mathcal{P}_1(E)$ which turns the pair $\mathcal{W}_1 := (\mathcal{P}_1(E), \mathrm{W}_1)$ into a complete, separable metric space (Villani, 2008, Ch 6).

### 5.1 Lipschitz Contractions: Bounded Case

We now derive a Lipschitz contraction estimate for the map $\mu \mapsto \mu \mathbf{A}_\mu$ on the metric space $(\mathcal{P}_1(E), \mathrm{W}_1)$ via an inequality of the form:

$$\sup_{\mu \neq \nu} \mathrm{W}_1(\mu \mathbf{A}_\mu, \nu \mathbf{A}_\nu) \leq \tau(\mathbf{A}) \mathrm{W}_1(\mu, \nu)$$

for some constant $\tau(\mathbf{A})$ to be determined. In this Section, we make the additional assumption.

**Assumption 2.** $E \subset \mathbb{R}^d$ is compact.

We will estimate the Wasserstein contraction coefficient defined below.

**Definition 7** (Wasserstein Contraction Coefficient). *Let $\Phi : \mathcal{P}_1(E) \to \mathcal{P}_1(E)$ be a (possibly nonlinear) mapping. We define the **Wasserstein contraction coefficient** by*

$$\tau(\Phi) := \sup_{\mu \neq \nu} \frac{\mathrm{W}_1(\Phi(\mu), \Phi(\nu))}{\mathrm{W}_1(\mu, \nu)}.$$

**Remark 4.** *This definition is a natural extension of two concepts from applied probability: it is the generalization of the total variation contraction coefficient studied in Del Moral (2004) for nonlinear Markov operators to the 1-Wasserstein distance; it is also the extension of the generalized ergodic coefficient from Rudolf et al. (2018) to nonlinear Markov operators.*

Also, for $f : E \to \mathbb{R}$, the Lipschitz semi-norm is $\|f\|_{Lip} := \sup_{x \neq y} |f(x) - f(y)|/d(x, y)$. For a function $G$ of two variables, $G : E \times E \to \mathbb{R}$, set:

$$\|G\|_{Lip,\infty} := \sup_{x \in E} \|G(\bullet, x)\|_{Lip} \qquad \qquad \|G\|_{\infty, Lip} := \sup_{x \in E} \|G(x, \bullet)\|_{Lip}.$$

**Theorem 1.** *Let $E \subset \mathbb{R}^d$ be compact and convex, and let $\mathbf{A}$ be the attention kernel from Definition 6 with $G$ an interaction potential s.t. $G(x, y) \geq \epsilon(G) > 0$, $\|G\|_{Lip,\infty} < \infty$ and $\|G\|_{\infty, Lip} < \infty$. Then the 1-Wasserstein contraction coefficient $\tau(\mathbf{A})$ of $\mathbf{A}$ considered as a mapping $\mathcal{P}(E) \to \mathcal{P}(E)$ via $\mathbf{A} : \mu \mapsto \mu \mathbf{A}_\mu$ satisfies*

$$\tau(\mathbf{A}) \leq \tau(\Pi) \tau(\Psi_G) \tau(L)$$

*where $\tau(\Psi_G) = \frac{2(\|G\|_{Lip,\infty} + \|G\|_{\infty, Lip}) \mathrm{diam}(E)}{\epsilon(G)}$ and $\tau(\Pi) = d$. Additionally, if $L(x, \mathrm{d}y) = \delta_{\ell(x)}(\mathrm{d}y)$, then $\tau(L) = \|\ell\|_{Lip}$.*

*Proof.* See Appendix B. $\qquad \qquad \square$

**Corollary 1.** *Let $K = \{k_1, \ldots, k_N\} \subset E \subset \mathbb{R}^d$ and $V = \{v_1, \ldots, v_N\} \subset E \subset \mathbb{R}^d$ and the attention function $\mathrm{Attention}(\bullet, K, V)$ be as in the original definition of attention from Bahdanau et al. (2014), Definition 1. Assume that the components of $\mathrm{Attention}(\bullet, K, V)$ satisfy Theorem 1. Then the mapping*

$$q \mapsto \mathrm{Attention}(q, K, V)$$

*is Lipschitz continuous as a mapping from $\mathbb{R}^d \to \mathbb{R}^d$ with the Euclidean distance, and moreover*

$$\|\mathrm{Attention}(q_1, K, V) - \mathrm{Attention}(q_2, K, V)\|_2 \leq d^{3/2} \cdot \|\ell\|_{Lip} \cdot \frac{2\|G\|_{Lip,\infty} \mathrm{diam}(E)}{\epsilon(G)} \cdot \|q_1 - q_2\|_2$$

*Proof.* Using elements from the proof of Theorem 1 in Appendix B, we have:

$$\|\text{Attention}(q_1, K, V) - \text{Attention}(q_2, K, V)\|_1 = \mathbb{W}_1(\delta_{q_1}\mathbf{A}_{m(K)}, \delta_{q_2}\mathbf{A}_{m(K)})$$

$$\leq d \cdot \|\ell\|_{Lip} \cdot \frac{2\|G\|_{Lip,\infty}\text{diam}(E)}{\epsilon(G)} \cdot \mathbb{W}_1(\delta_{q_1}, \delta_{q_2})$$

$$= d^{3/2} \cdot \|\ell\|_{Lip} \cdot \frac{2\|G\|_{Lip,\infty}\text{diam}(E)}{\epsilon(G)} \cdot \|q_1 - q_2\|_2$$

using $\|x\|_2 \leq \|x\|_1 \leq \sqrt{d}\|x\|_2$ and that $\|\ell\|_{Lip} = 1$ for vanilla self-attention where $\ell(x) = x$. $\qquad\square$

## 5.2 Lipschitz Contractions: Unbounded Case

The results of Section 5.1 depend on the boundedness of the representation space $E$. While this is sufficient to provide rather general estimates on the Lipschitz coefficient for attention that are verified by reasonable choices for $G$ and $L$, it is natural to question if it is *necessary*. As we will discuss below, the answer is affirmative, at least in full generality.

In recent work by Kim et al. (2021), the authors investigate Lipschitz constants for self-attention on $X = \{x_1, \ldots, x_N\}$ as a mapping from $\mathbb{R}^{d \times N} \to \mathbb{R}^{d \times N}$ without assuming $E$ is bounded. They show that, for the case of $G(x, y) = \exp\langle x, y\rangle$ on the whole of $\mathbb{R}^d$, attention is not Lipschitz by proving that the norm of the Jacobian is unbounded (Kim et al. (2021) Theorem 3.1). The authors then show that using instead the interaction potential $G(x, y) = \exp(-\|x - y\|_2^2/\sqrt{d})$ leads to a Lipschitz bound independent of $\text{diam}(E)$ (Kim et al. (2021) Theorem 3.2). They also provide empirical evidence that this potential function does not severely degrade performance.

We provide below an analysis of a similar Gaussian interaction potential $G(x, y) = \exp(-\|x - y\|_2^2)$ as in Kim et al. (2021)[2] for unbounded $E = \mathbb{R}^d$. We are able to use a set of tools and approach similar to those from Section 5.1 but exchange the boundedness assumption on $E$ for exponential decay of $G(x, y)$ and $\|\nabla G(x, y)\|$ as $\|x - y\|_2 \to \infty$. The proofs are in Appendix C.

**Theorem 2.** *Let $E = \mathbb{R}^d$ and suppose $X = \{x_1, \ldots, x_N\}, Y = \{y_1, \ldots, y_M\} \subset \mathbb{R}^d$. Let $G(x, y) = \exp(-\|x - y\|_2^2)$ and $\Pi$ be the usual projection onto $\mathcal{P}_\delta(\mathbb{R}^d)$. Then for $\mu = m(X)$ and $\nu = m(Y)$,*

$$\mathbb{W}_1(\mu\mathbf{A}_\mu, \nu\mathbf{A}_\nu) \leq 2\tau(\Pi)\tau(L)\left[\|G\|_\infty + \sqrt{d} + 2 + \sqrt{d}\sqrt{\ln(\min(N, M)) + \frac{1}{2e}}\|G\|_{Lip}\right]\mathbb{W}_1(\mu, \nu).$$

Theorem 2 provides an alternate path to the Lipschitz constant of self-attention compared to methods based on computing Jacobians (Kim et al., 2021). In particular, Theorem 2 applies to sequences of tokens of various lengths and allows for studying the effect of perturbing a sequence by e.g. removing a given word, or negating a sentence, which is out of immediate reach for Jacobian-based techniques. Finally, we can recover a bound for sequences of equal lengths:

**Corollary 2.** *Applying Theorem 2 to the case of $N = M$ gives:*

$$\mathbb{W}_1(\mu\mathbf{A}_\mu, \nu\mathbf{A}_\nu) \leq 2d\tau(L)\left[\sqrt{d}\sqrt{\ln N + \frac{1}{2e}}\|G\|_{Lip} + \|G\|_\infty + \sqrt{d} + 2\right]\mathbb{W}_1(\mu, \nu).$$

**Optimality of Lipschitz Estimates.** First, let us consider the $\mathcal{O}(\text{diam}(E)/\varepsilon(G))$ dependence in Theorem 1 in the case of bounded $E$ (recall $\varepsilon(G) := \inf_{x \in E} G(x)$). While in practice these values may lead to large bounds, we do not believe they indicate obvious inefficiencies in our technique. Indeed, we cannot simultaneously relax the finiteness of $\text{diam}(E)$ and $\varepsilon(G)$ in the general case: dot-product attention is a non-pathological counterexample (Kim et al., 2021). We believe it is likely than one cannot relax $\text{diam}(E) < \infty$ in the general case either, but we will study this in future work.

---

[2]We chose the un-parameterized potential for simplicity, we see no reason our framework would not extend to the parameterized case as well.

Second, for a *trained* attention network, $\text{diam}(E) < \infty$ and $\varepsilon(G) > 0$ are automatically satisfied, so these estimates can be used to study the very common use-case of pre-trained models. A potentially useful consequence of these estimates is an easy "knob" to control the regularity of an attention model by controlling $\text{diam}(E)$ (e.g. by projecting on a ball of fixed radius).

Finally, the appearance of an additional factor of $\sqrt{d}$ is the cost we pay for using $\mathbb{W}_1$, which relies on the $\ell_1$ metric in $\mathbb{R}^d$, to provide $\ell_2$- Lipschitz bounds. This is likely not optimal; it may be possible to derive a similar result with the 2-Wasserstein which would likely enjoy the good properties of the Wasserstein distance without the penalty of $\sqrt{d}$ (since $\|x - y\|_2 = \mathbb{W}_2(\delta_x, \delta_y)$) but it will not use the Lipschitz duality we have exploited in this paper which is specific to $\mathbb{W}_1$.

## 6 Discussion

In this section, we will apply the analysis developed above to discuss some consequences of regularity. Firstly, we will show that a common use of attention (called "cross attention") is also (Lipschitz) continuous w.r.t. the input keys. We then highlight cases where regularity either helps or hurts performance on various tasks. Finally, we discuss the implications of regularity on the invertibility of self-attention networks, and the case of infinitely deep, weight-tied self-attention networks.

### 6.1 Cross Attention is Continuous w.r.t. Keys

Although we have been primarily interested in the question of self-attention so far, the tools we have developed also apply to other uses of attention. One common example is **cross-attention**, i.e. when the keys and values are the same, but the queries can be different $q, X \mapsto \text{Attention}(q, X, X)$. This is used in practice when one wants to construct a context-specific representation of $q$ in the same "semantic space" as $X$ (hence $X$ provides the values). Notably, this is used in the seqence2sequence (or encoder-decoder) architecture (Sutskever et al., 2014), where $X$ represents the encoded sequence and $q$ represents the current element being decoded, see e.g. Bahdanau et al. (2014); Vaswani et al. (2017). There has also been recent work in proposing new attention-based architectures in which cross-attention plays a critical role such as the Perceiver (Jaegle et al., 2021; Alayrac et al., 2022).

Our framework shows that the resulting representation is Lipschitz continuous w.r.t. the output semantic space $X$. Note that this result highlights the flexibility of our results: two input spaces $X, Y$ need not even have the same length!

**Proposition 3.** *Suppose that $q \in \mathbb{R}^{d_q}$ $X := \{x_1, \ldots, x_N\} \subset \mathbb{R}^{d_k}$ and $Y := \{y_1, \ldots, y_{N'}\} \subset \mathbb{R}^{d_k}$ are sets of vectors for $N, N' \in \mathbb{N}$, and suppose that the assumptions of Theorem 1 hold. Then*

$$\|\text{Attention}(q, X, X) - \text{Attention}(q, Y, Y)\|_2 \le d \cdot \tau(L) \frac{2\|G(q, \bullet)\|_{Lip}\text{diam}(E)}{\varepsilon(G)} \cdot \mathbb{W}_1(m(X), m(Y))$$

*Proof.* We can adapt an argument from the proof of Theorem 1. Firstly, for simplicity write $\mu := m(X), \nu := m(Y)$ and note that

$$\|\text{Attention}(q, X, X) - \text{Attention}(q, Y, Y)\|_2 \le \|\text{Attention}(q, X, X) - \text{Attention}(q, Y, Y)\|_1$$
$$= \mathbb{W}_1(\delta_q \mathbf{A}_\mu, \delta_q \mathbf{A}_\nu).$$

Then by Proposition 5

$$\mathbb{W}_1(\delta_q \mathbf{A}_\mu, \delta_q \mathbf{A}_\nu) = \mathbb{W}_1(\delta_q \Pi[\Psi_{G(\bullet, \bullet)}(\mu)L], \delta_q \Pi[\Psi_{G(\bullet, \bullet)}(\nu)L])$$
$$= \mathbb{W}_1(\Pi[\Psi_{G(q, \bullet)}(\mu)L], \Pi[\Psi_{G(q, \bullet)}(\nu)L]) \le \tau_1(\Pi)\tau_1(L)\mathbb{W}_1(\Psi_{G(q, \bullet)}(\mu), \Psi_{G(q, \bullet)}(\nu))$$
$$\le \tau_1(\Pi)\tau_1(L)\frac{2\|G(q, \bullet)\|_{Lip}\text{diam}(E)}{\varepsilon(G)}\mathbb{W}_1(\mu, \nu) = d \cdot \tau_1(L)\frac{2\|G(q, \bullet)\|_{Lip}\text{diam}(E)}{\varepsilon(G)}\mathbb{W}_1(\mu, \nu).$$

$\square$

In the case that $|X| = |Y| = N$, we can obtain an explicit formula for $\mathbb{W}_1(m(X), m(Y))$ (see e.g. Bobkov & Ledoux (2014), Lemma 4.2):

$$\mathbb{W}_1(m(X), m(Y)) = \inf_{\sigma \in \Sigma(N)} \frac{1}{N} \sum_{i=1}^{N} \|x_s - y_{\sigma(s)}\|_1$$

where $x_s \in X$, $y_s \in Y$ and $\Sigma(m)$ is the set of permutations on $m$ elements.

## 6.2 Robustness and Perturbations

**Robustness to noisy inputs and adversarial examples.** Robustness and Lipschitzness are very tied concepts (Bubeck & Sellke, 2021). One effect of the smoothness of attention is that the representations it produces are "robust to errors" to a certain degree. For instance, in the encoder-decoder setup mentioned above, if the outputs of an encoder are incorrect or noisy, an attention-based decoder still has a chance of performing adequately. This robustness has been used in Anderson et al. (2020) to operate self-attention transformer models on reduced-size vocabularies by hashing, where the model must be robust to hash collisions of the larger original vocabulary. The authors of that paper compare this robustness to error correcting output codes (Dietterich & Bakiri, 1994; Berger, 1999). Our framework provides a potential mathematical basis for this phenomenon in transformers.

**Negated Sentences.** This robustness is not always desirable, however. Indeed, our regularity results may also explain some recent observations on the behavior of deep language models with respect to negation. Table 4 of Kassner & Schütze (2019) shows that negated sentences are often given identical predictions to the original ones: for instance, both "A beagle is a type of [MASK]" and "A beagle is **not** a type of [MASK]" get a prediction of "dog". To address the issue, Hosseini et al. (2021) had to regularize the language model using an "unlikelihood" objective on generic negated sentences.

One hypothesis for why this phenomenon occurs without specific regularization is a "regularity mismatch" between the input space and the output space of the model. On one hand, negation is a type of perturbation in "token space" that drastically changes the semantic content of the sentence, i.e. it is highly irregular. On the other hand, our analysis — specifically, Prop. 3 — suggests that the resulting embeddings will not change "too much" in response to this perturbation. If the embeddings are close with and without negation, i.e. the model is "too smooth" w.r.t. perturbations in token space, the scoring network (often a linear classifier) will not be able to distinguish between the resulting embeddings and the model will fail.

Our modelling could potentially be used to derive predictions of the distance between a self-attention networks' contextual embeddings *as a function of the context* (e.g. for sentences with and without a "not") to test this hypothesis. Moreover, it could even potentially be used to design better model components (e.g. input embedding spaces) that reduce this "regularity mismatch" for specific perturbations that are highly irregular. See Appendix D for a preliminary experiment illustrating this effect with a pretrained BERT model Devlin et al. (2018).

## 6.3 Invertible & Infinite Depth Transformers

Finally, let us briefly mention two important consequences of the Lipschitz regularity of attention: invertibility (also studied empirically in Kim et al. (2021)) and infinite-depth attention networks.

**Invertibility.** Firstly, as noted in Behrmann et al. (2019), a sufficient condition for invertibility of a residual network of the form $F(x) = F_L \circ \cdots \circ F_1(x)$ where each residual block $F_\ell$ has the form

$$F_\ell(x) = x + g_\ell(x)$$

is the Lipschitz condition $\|g_\ell\|_{Lip} < 1$ for $\ell = 1, \ldots, L$. The self-attention Transformer from Vaswani et al. (2017) uses self-attention exactly this way, where $g_\ell(X) = \text{SelfAttention}(X)$ (it also uses a feedforward residual block). Therefore, our results provide sufficient conditions for a deep self-attention transformer to be invertible. Note that this general conclusion was also used in Kim et al. (2021). Moreover, our analysis

could be applied to the scaled dot product potential function (Vaswani et al., 2017) by enforcing that the input representations come from a bounded subset of $\mathbb{R}^d$. This is in contrast with the work of Kim et al. (2021), whose Lipschitz constants only apply to the Gaussian interaction potential.

**Infinitely-Deep Attention Models.** In the opposite direction of invertibility, infinitely-deep models have recently been studied in the context of "deep equilibrium models" (Bai et al., 2019). The authors study representations defined as fixed points

$$H^* = f_\theta(H^*; X) \tag{4}$$

where $f_\theta$ is an *input-injected* nonlinear function and $H^* = \{h_1^*, \ldots, h_N^*\}$ is a collection of hidden representations for the inputs $X = \{x_1, \ldots, x_N\}$. Here *input-injected* means $f_\theta$ includes a (possibly parameterized) skip connection $s_\theta$ from the inputs to the hidden representations of the form

$$f_\theta(H; X) = g_{\theta_1}(H + s_{\theta_2}(X)).$$

Note that the Banach Fixed Point Theorem provides a sufficient condition for the existence of $H^*$: the mapping $H \mapsto f_\theta(H; X)$ has Lipschitz constant $< 1$.

In Bai et al. (2019), the authors note that the model in (4) includes the Universal Transformer model (Dehghani et al., 2018), albeit with the minor modification of including an "input injection" connection. In this situation, $f_\theta$ is self-attention so we can apply our theory to obtain sufficient conditions on the existence of $H^*$ from Theorem 1 or Theorem 2 depending on the type of attention used. We didn't find an existence result such as this in Bai et al. (2019).

In light of our results, we understand why the input injection is important: it produces a *data-dependent* fixed point. If (4) had no the skip-connection (and no way to parameterize $f_\theta$ in-terms of $X$), the fixed point $H^*$ would not depend on the inputs and therefore be of questionable usefulness.

## 7 Conclusion

In this paper, we have studied the regularity of attention. In particular, we have shown that attention is Lipschitz continuous under various assumptions, and provided estimates of the Lipschitz constant. To do so, we have introduced an alternate, but equivalent, modelling paradigm for attention based on measure theory and integral operators. We then assessed the impact of these regularity results on study practical applications of attention, including cross-attention; robustness and token-level perturbations in NLP; and sophisticated extensions to the transformer architecture.

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

# A The Transformer

In this section, we show how to extend the measure-theoretic model of self-attention described in the main text to the full Transformer encoder architecture (Vaswani et al., 2017)[3]. This is a straightforward application of the techniques from the main text. For our purpose, we work with the model

$$\text{Transformer}(X) = \text{FFN} \circ \text{MultiHeadSelfAttention}(X), \tag{5}$$

where $X = \{x_1, \ldots, x_N\} \subset \mathbb{R}^d$ and FFN represents a feedforward neural network. To incorporate this into our formalism above, first set $\widetilde{a}(x, y) = x^T y / \sqrt{d}$. We can model a single head of the Transformer using the attention kernel from Definition 6 with:

$$a(x, y) = \widetilde{a}\left(W^Q x, W^K y\right), \qquad L(k, \mathrm{d}v) = \delta_{W^V k}(\mathrm{d}v),$$

where $W^Q, W^K, W^V$ are matrices in $\mathbb{R}^{d' \times d}$ where $d'$ can possibly be a different dimension than $d$. To model multi-headed attention, we note that multi-headedness amounts to processing independent copies of the data $X$ and combining them with concatenation and matrix multiplication. The "concat-and-matmult" operation can be written as

$$\begin{bmatrix} x_i^1 & \cdots & x_i^H \end{bmatrix} \begin{bmatrix} W_1^O \\ \vdots \\ W_H^O \end{bmatrix} = x_i^1 W_1^O + \cdots + x_i^H W_H^O,$$

where each $W_h^O \in \mathbb{R}^{d' \times d}$. Hence, letting $\mathbf{O}^h(x, \mathrm{d}y) := \delta_{x W_h^O \cdot H}(\mathrm{d}y)$, where we have multiplied by the scalar $H$, and introducing the mixture kernel

$$\widehat{\mathbf{M}} := \frac{1}{H} \sum_{h=1}^{H} \mathbf{A}^h \mathbf{O}^h,$$

where each $h$ parameterizes its own collection of projection matrices and attention head $\mathbf{A}^h$, we can define the multi-headed attention attention kernel as

$$\mathbf{M} := \Pi \circ \widehat{\mathbf{M}}, \qquad \mathbf{M}_\mu(x, \mathrm{d}y) = \Pi(\widehat{\mathbf{M}}_\mu(x, \bullet))(\mathrm{d}y).$$

Finally, letting $f : E \to E$ be the FFN in 5 and defining the FFN kernel as $\mathbf{F}(x, \mathrm{d}y) = \delta_{f(x)}(\mathrm{d}y)$, we see that $\mathbf{T} := \mathbf{MF}$ implements the self-attention transformer as nonlinear measure transport.

**Proposition 4** (prop. 2 restated). *Let $X = \{x_1, \ldots, x_N\} \subset \mathbb{R}^d$ be a collection of inputs. The nonlinear Markov transport equation $\delta_{x_i} \mapsto \delta_{x_i} \mathbf{T}_{m(X)}$ implements the self-attention Transformer.*

*Proof.* Given the discussion about standard attention, the only new element to be checked is the multi-headed attention kernel. Consider a fixed $X$, then

$$m(X) \mathbf{M}_{m(X)}(\mathrm{d}y) = \frac{1}{N} \sum_{i=1}^{N} \int \delta_{x_i} \mathbf{M}_{m(X)}(x, \mathrm{d}y) = \frac{1}{N} \sum_{i=1}^{N} \mathbf{M}_{m(X)}(x_i, \mathrm{d}y)$$

Hence considering a single $x_i$, we see that

$$\mathbf{M}_{m(X)}(x_i, \mathrm{d}y) = \Pi\left(\widehat{\mathbf{M}}_{m(X)}(x_i, \bullet)\right)(\mathrm{d}y).$$

The inner kernel is

$$\widehat{\mathbf{M}}_{m(X)}(x_i, \mathrm{d}y) = \frac{1}{H} \sum_{h=1}^{H} \int \mathbf{A}_{m(X)}^h(x_i, \mathrm{d}z) \mathbf{O}^h(z, \mathrm{d}y) = \frac{1}{H} \sum_{h=1}^{H} \int \mathbf{A}_{m(X)}^h(x_i, \mathrm{d}z) \delta_{z W_h^O \cdot H}(\mathrm{d}y).$$

---

[3]Technically, the Transformer also contains layer normalization and residual connections, which we do not treat here.

The measure $\mathbf{A}^h_{m(X)}(x_i, \mathrm{d}z)$ is a delta-measure concentrated on the point

$$\sum_{j=1}^N \frac{\exp[\widetilde{a}(W_h^Q x_i, W_h^K x_j]}{\sum_{p=1}^N \exp[\widetilde{a}(W_h^Q x_i, W_h^K x_k)]} W_h^V x_j = \text{MultiHeadSelfAttention}(x_i, X, X)_h =: y_i^h$$

hence

$$\frac{1}{H} \sum_{h=1}^H \int \mathbf{A}^h_{m(X)}(x_i, \mathrm{d}z)\delta_{zW_h^O \cdot H}(\mathrm{d}y) = \frac{1}{H} \sum_{h=1}^H \int \delta_{y_i^h}(\mathrm{d}z)\delta_{zW_h^O \cdot H}(\mathrm{d}y) = \frac{1}{H} \sum_{h=1}^H \delta_{y_i^h W_h^O \cdot H}(\mathrm{d}y).$$

Finally, applying the mapping $\Pi$ we get a measure that is concentrated on the point

$$\int_E \frac{1}{H} \sum_{h=1}^H \delta_{y_i^h W_h^O \cdot H}(\mathrm{d}y)y = \frac{1}{H} \sum_{h=1}^H y_i^h W_h^O \cdot H = \begin{bmatrix} y_i^1 & \cdots & y_i^H \end{bmatrix} \begin{bmatrix} W_1^O \\ \vdots \\ W_H^O \end{bmatrix}$$

$$= \text{MultiHeadSelfAttention}(x_i, X, X),$$

which concludes the proof. $\qquad\square$

## B    Proofs From Section 5.1

**Proposition 5.** *Suppose $\mu, \nu \in \mathcal{P}_1(E)$ and $G : E \times E \to \mathbb{R}$, $G(x, y) \geq \epsilon(G) > 0$ is an interaction potential s.t. $\|G\|_{\infty,Lip} < \infty$ and $\|G\|_{Lip,\infty} < \infty$. Then, $\forall x, y \in E$:*

$$\mathbb{W}_1(\Psi_{G(x,\bullet)}(\mu), \Psi_{G(y,\bullet)}(\mu)) \leq 2\frac{\|G\|_{Lip,\infty}\text{diam}(E)}{\epsilon(G)} \cdot d(x,y),$$

$$\mathbb{W}_1(\Psi_{G(x,\bullet)}(\mu), \Psi_{G(x,\bullet)}(\nu)) \leq \frac{2\|G\|_{\infty,Lip}\text{diam}(E)}{\epsilon(G)} \cdot \mathbb{W}_1(\mu, \nu).$$

*Proof.* For the first inequality, let $f$ be any 1-Lipschitz function, and $x, y \in E$. We have:

$$|\Psi_{G(x,\bullet)}(\mu)(f) - \Psi_{G(y,\bullet)}(\mu)(f)| = \left| \int \frac{G(x,z)f(z)}{\mu(G(x,\bullet))} - \frac{G(y,z)f(z)}{\mu(G(y,\bullet))} \mu(dz) \right|$$

$$\leq \left| \int \frac{G(x,z)f(z)}{\mu(G(x,\bullet))} - \frac{G(x,z)f(z)}{\mu(G(y,\bullet))} \mu(dz) \right|$$

$$+ \left| \int \frac{G(x,z)f(z)}{\mu(G(y,\bullet))} - \frac{G(y,z)f(z)}{\mu(G(y,\bullet))} \mu(dz) \right|.$$

Let us bound the first term:

$$\left| \int \frac{G(x,z)f(z)}{\mu(G(x,\bullet))} - \frac{G(x,z)f(z)}{\mu(G(y,\bullet))} \mu(dz) \right| \leq \frac{|\mu(G(x,\bullet)) - \mu(G(y,\bullet))|}{\mu(G(x,\bullet))\mu(G(y,\bullet))} \int G(x,z)|f(z)|\mu(dz)$$

$$\leq \frac{\int |G(x,z) - G(y,z)|\mu(dz)}{\mu(G(x,\bullet))\epsilon(G)}\mu(G(x,\bullet))\|f\|_\infty$$

$$\leq \frac{\int |G(x,z) - G(y,z)|\mu(dz)\|f\|_\infty}{\epsilon(G)d(x,y)}d(x,y)$$

$$\leq \frac{\|G\|_{Lip,\infty}\|f\|_\infty}{\epsilon(G)}d(x,y).$$

Let us now bound the second term:

$$\left| \int \frac{G(x,z)f(z)}{\mu(G(y,\bullet))} - \frac{G(y,z)f(z)}{\mu(G(y,\bullet))} \mu(dz) \right| \leq \frac{\|f\|_\infty}{\epsilon(G)} \int |G(x,z) - G(y,z)|\mu(dz)$$

$$\leq \frac{\|G\|_{Lip,\infty}\|f\|_\infty}{\epsilon(G)}d(x,y).$$

Using the fact that $\Psi_{G(x,\bullet)}(\mu)(\bar{f}) = \Psi_{G(y,\bullet)}(\mu)(\bar{f})$ for any constant function $\bar{f}$, we can subtract from $f$ any constant without changing the value of $|\Psi_{G(x,\bullet)}(\mu)(f) - \Psi_{G(y,\bullet)}(\mu)(f)|$. This allows us to assume without loss of generality that $\|f\|_\infty \leq \operatorname{diam}(E)$ (picking an arbitrary $x \in E$, we have $\forall y \in E$, $|f(y) - f(x)| \leq |y - x| \|f\|_{Lip} \leq \operatorname{diam}(E)$). Combining everything, we get:

$$|\Psi_{G(x,\bullet)}(\mu)(f) - \Psi_{G(y,\bullet)}(\mu)(f)| \leq 2\frac{\|G\|_{Lip,\infty}\operatorname{diam}(E)}{\epsilon(G)}d(x,y).$$

Taking the supremum over 1-Lipschitz functions $f$ concludes the first part of the proof.

Let us now prove the second inequality. Similarly, let $f$ be any 1-Lipschitz function, and $\mu, \nu$ two compactly supported distributions on $(E, \mathcal{E})$. We use the notation $G(z) := G(x, z)$ for this part because $x$ is fixed. We have:

$$
\begin{aligned}
|\Psi_G(\mu)(f) - \Psi_G(\nu)(f)| &= \left| \int \frac{G(z)f(z)}{\mu(G)}\mu(dz) - \int \frac{G(z)f(z)}{\nu(G)}\nu(dz) \right| \\
&\leq \left| \int \frac{G(z)f(z)}{\mu(G)}\mu(dz) - \int \frac{G(z)f(z)}{\nu(G)}\mu(dz) \right| \\
&\quad + \left| \int \frac{G(z)f(z)}{\nu(G)}\mu(dz) - \int \frac{G(z)f(z)}{\nu(G)}\nu(dz) \right|.
\end{aligned}
$$

Let us bound the first term:

$$
\begin{aligned}
\left| \int \left( \frac{G(z)f(z)}{\mu(G)} - \frac{G(z)f(z)}{\nu(G)} \right) \mu(dz) \right| &\leq \frac{|\mu(G) - \nu(G)|}{\mu(G)\nu(G)} \int G(z)|f(z)|\mu(dz) \\
&\leq \frac{\|G\|_{Lip}\mathrm{W}_1(\mu,\nu)}{\mu(G)\epsilon(G)}\mu(G)\|f\|_\infty \\
&\leq \frac{\|G\|_{Lip}\|f\|_\infty}{\epsilon(G)}\mathrm{W}_1(\mu,\nu).
\end{aligned}
$$

Let us now bound the second term:

$$
\begin{aligned}
\left| \int \frac{G(z)f(z)}{\nu(G)}\mu(dz) - \int \frac{G(z)f(z)}{\nu(G)}\nu(dz) \right| &\leq \frac{\|f\|_\infty}{\nu(G)} \left| \int G(z)\mu(dz) - \int G(z)\nu(dz) \right| \\
&\leq \frac{\|G\|_{Lip}\|f\|_\infty}{\epsilon(G)}\mathrm{W}_1(\mu,\nu).
\end{aligned}
$$

Using the same reasoning as above, we can assume without loss of generality that $\|f\|_\infty \leq \operatorname{diam}(E)$, which gives:

$$|\Psi_G(\mu)(f) - \Psi_G(\nu)(f)| \leq 2\frac{\|G\|_{Lip}\operatorname{diam}(E)}{\epsilon(G)}\mathrm{W}_1(\mu,\nu).$$

Taking the supremum over all 1-Lipschitz functions $f$ concludes the proof. $\qquad\square$

**Proposition 6.** *Suppose that $\Pi : \mathcal{P}(E) \to \mathcal{P}_\delta(E)$ is the measure projection $\mu \mapsto \delta_{\overline{\mu}}$, where $\overline{\mu} = \int x\mu(\mathrm{d}x)$. Then, for $\mu, \nu \in \mathcal{P}_1(E)$, $\mathrm{W}_1(\Pi(\mu), \Pi(\nu)) \leq d \cdot \mathrm{W}_1(\mu, \nu)$.*

*Proof.* Denote by $\pi_i : E \to \mathbb{R}$ the canonical projection onto the $i$-th coordinate of $E \subset \mathbb{R}^d$, and let $x_i := \pi_i(x)$. Moreover, denote $F(x) = x$, remarking that $\mu(F) = \int F(x)\mu(\mathrm{d}x) = \int x\mu(\mathrm{d}x) = \overline{\mu}$. Then

$$
\begin{aligned}
\mathbb{W}_1(\Pi(\mu), \Pi(\nu)) &= \mathbb{W}_1(\delta_{\mu(F)}, \delta_{\nu(F)}) \\
&= \|\mu(F) - \nu(F)\|_1 \\
&= \sum_{i=1}^{d} |\mu(F)_i - \nu(F)_i| \\
&= \sum_{i=1}^{d} |\mu(\pi_i \circ F) - \nu(\pi_i \circ F)| \\
&\leq d \cdot \max_{i=1,\ldots,d} \{|\mu(\pi_i \circ F) - \nu(\pi_i \circ F)|\} \\
&\leq d \cdot \sup_{f \in Lip(1)} |\mu(f) - \nu(f)| \\
&= d \cdot \mathbb{W}_1(\mu, \nu)
\end{aligned}
$$

since $\pi_i \circ F \in Lip(1)$ for $i = 1, \ldots, d$. $\qquad\square$

**Proposition 7.** *Suppose $L : E \times \mathcal{E} \to [0, 1]$ is a lookup kernel implementing a deterministic lookup function $\ell : E \to E$, (i.e. $L(x, \mathrm{d}y) = \delta_{\ell(x)}(\mathrm{d}y)$) and suppose that $\ell$ is $K_\ell$-Lipschitz in the 1-norm, then $\mathbb{W}_1(\mu L, \gamma L) \leq K_\ell \mathbb{W}_1(\mu, \gamma)$.*

*Proof.*

$$
\begin{aligned}
\mathbb{W}_1(\mu L, \gamma L) &= \sup_{f \in Lip(1)} \left| \int f(x)\mu L(\mathrm{d}x) - \int f(y)\gamma L(\mathrm{d}y) \right| \\
&= \sup_{f \in Lip(1)} \left| \int f(x) \int \mu(\mathrm{d}z)L(z, \mathrm{d}x) - \int f(y) \int \gamma(\mathrm{d}z)L(z, \mathrm{d}y) \right| \\
&= \sup_{f \in Lip(1)} \left| \iint f(x)L(z, \mathrm{d}x)\mu(\mathrm{d}z) - \iint f(y)L(z, \mathrm{d}y)\gamma(\mathrm{d}z) \right| \\
&= \sup_{f \in Lip(1)} \left| \iint f(x)\delta_{\ell(z)}(\mathrm{d}x)\mu(\mathrm{d}z) - \iint f(y)\delta_{\ell(z)}(\mathrm{d}y)\gamma(\mathrm{d}z) \right| \\
&= \sup_{f \in Lip(1)} \left| \int f \circ \ell(z)\mu(\mathrm{d}z) - \int f \circ \ell(z)\gamma(\mathrm{d}z) \right|.
\end{aligned}
$$

Then since $\|f\|_{Lip} = 1$, we have $\|f \circ \ell\|_{Lip} \leq \|f\|_{Lip}\|\ell\|_{Lip} = K_\ell$. Hence, by our earlier estimation techniques:

$$
\begin{aligned}
\mathbb{W}_1(\mu L, \gamma L) &= \sup_{f \in Lip(1)} \left| \int f \circ \ell(\mathrm{d}z)\mu(\mathrm{d}z) - \int f \circ \ell(z)\gamma(\mathrm{d}z) \right| \\
&\leq K_\ell \sup_{g \in Lip(1)} \left| \int g(\mathrm{d}z)\mu(\mathrm{d}z) - \int g(z)\gamma(\mathrm{d}z) \right| = K_\ell \mathbb{W}_1(\mu, \gamma),
\end{aligned}
$$

which concludes the proof. $\qquad\square$

**Lemma 1.**  1. *Suppose that $\Phi, \Gamma : \mathcal{P}(E) \to \mathcal{P}(E)$ are (possibly nonlinear) mappings. Then*

$$
\tau(\Phi \circ \Gamma) \leq \tau(\Phi)\tau(\Gamma).
$$

2. *Suppose $K : E \times \mathcal{E} \to [0, 1]$ is an integral kernel. Then*

$$
\tau(K) = \sup_{x \neq y} \frac{\mathbb{W}_1(K(x, \bullet), K(y, \bullet))}{d(x, y)}.
$$

3. *Suppose $K_1, K_2 : E \times \mathcal{E} \to [0, 1]$ are two integral kernels and $\nu \in \mathcal{P}(E)$. Then:*

$$\mathbb{W}_1(\nu K_1, \nu K_2) \leq \int \nu(dx) \mathbb{W}_1(K_1(x, \bullet), K_2(x, \bullet)).$$

*Proof.*     1. This is a standard result on Lipschitz constants. We include it for completeness:

$$\begin{aligned}
\tau(\Phi \circ \Gamma) &= \sup_{\mu \neq \nu} \frac{\mathbb{W}_1(\Phi \circ \Gamma(\mu), \Phi \circ \Gamma(\nu))}{\mathbb{W}_1(\mu, \nu)} \\
&= \sup_{\mu \neq \nu} \frac{\mathbb{W}_1(\Phi \circ \Gamma(\mu), \Phi \circ \Gamma(\nu))}{\mathbb{W}_1(\Gamma(\mu), \Gamma(\nu))} \frac{\mathbb{W}_1(\Gamma(\mu), \Gamma(\nu))}{\mathbb{W}_1(\mu, \nu)} \\
&\leq \sup_{\eta \neq \gamma} \frac{\mathbb{W}_1(\Phi(\eta), \Phi(\gamma))}{\mathbb{W}_1(\eta, \gamma)} \cdot \sup_{\mu \neq \nu} \frac{\mathbb{W}_1(\Gamma(\mu), \Gamma(\nu))}{\mathbb{W}_1(\mu, \nu)} \\
&= \tau(\Phi)\tau(\Gamma).
\end{aligned}$$

2. Since $\mathbb{W}_1(\delta_x, \delta_y) = d(x, y)$ and $\delta_x K = K(x, \bullet)$ we have:

$$\sup_{x \neq y} \frac{\mathbb{W}_1(K(x, \bullet), K(y, \bullet))}{d(x, y)} = \sup_{\delta_x \neq \delta_y} \frac{\mathbb{W}_1(\delta_x K, \delta_y K)}{\mathbb{W}_1(\delta_x, \delta_y)} \leq \sup_{\mu \neq \nu} \frac{\mathbb{W}_1(\mu K, \nu K)}{\mathbb{W}_1(\mu, \nu)}.$$

For the reverse inequality,

$$\begin{aligned}
\mathbb{W}_1(\mu K, \nu K) &= \sup_{f \in Lip(1)} |\mu K(f) - \nu K(f)| \\
&= \sup_{f \in Lip(1)} |\mu(Kf) - \nu(Kf)| \\
&\leq \sup_{f \in Lip(1)} \|Kf\|_{Lip} \cdot \sup_{g \in Lip(1)} |\mu(g) - \nu(g)| \\
&\leq \sup_{f \in Lip(1)} \|Kf\|_{Lip} \cdot \mathbb{W}_1(\mu, \nu)
\end{aligned}$$

and

$$\begin{aligned}
\sup_{f \in Lip(1)} \|Kf\|_{Lip} &= \sup_{f \in Lip(1)} \sup_{x \neq y} \frac{\int K(x, dz) f(z) - \int K(y, dz) f(z)}{d(x, y)} \\
&= \sup_{f \in Lip(1)} \sup_{x \neq y} \frac{\int [K(x, dz) - K(y, dz)] f(z)}{d(x, y)} \\
&= \sup_{x \neq y} \frac{\mathbb{W}_1(K(x, \bullet), K(y, \bullet))}{d(x, y)}.
\end{aligned}$$

Dividing by $\mathbb{W}_1(\mu, \nu)$ gives us the reverse inequality and concludes the proof.

3. By definition, we have:

$$\begin{aligned}
\mathbb{W}_1(\nu K_1, \nu K_2) &= \sup_{f \in Lip(1)} |\nu K_1(f) - \nu K_1(f)| \\
&= \sup_{f \in Lip(1)} \left| \iint \nu(dx) K_1(x, dy) f(y) - \iint \nu(dx) K_2(x, dy) f(y) \right| \\
&\leq \sup_{f \in Lip(1)} \int \nu(dx) \left| \int K_1(x, dy) f(y) - K_2(x, dy) f(y) \right| \\
&\leq \int \nu(dx) \mathbb{W}_1(K_1(x, \bullet), K_2(x, \bullet)).
\end{aligned}$$

$\square$

Using Propositions 5, 6 and 7 and Lemma 1, we can prove Theorem 1.

**Theorem 1.** *Let $E \subset \mathbb{R}^d$ be compact and convex, and let $\mathbf{A}$ be the attention kernel from Definition 6 with $G$ an interaction potential s.t. $G(x,y) \geq \epsilon(G) > 0$, $\|G\|_{Lip,\infty} < \infty$ and $\|G\|_{\infty,Lip} < \infty$. Then the 1-Wasserstein contraction coefficient $\tau(\mathbf{A})$ of $\mathbf{A}$ considered as a mapping $\mathcal{P}(E) \to \mathcal{P}(E)$ via $\mathbf{A} : \mu \mapsto \mu \mathbf{A}_\mu$ satisfies*

$$\tau(\mathbf{A}) \leq \tau(\Pi)\tau(\Psi_G)\tau(L)$$

*where $\tau(\Psi_G) = \frac{2(\|G\|_{Lip,\infty} + \|G\|_{\infty,Lip})\mathrm{diam}(E)}{\epsilon(G)}$ and $\tau(\Pi) = d$. Additionally, if $L(x, \mathrm{d}y) = \delta_{\ell(x)}(\mathrm{d}y)$, then $\tau(L) = \|\ell\|_{Lip}$.*

*Proof.* We want to bound $\sup_{\mu \neq \nu} \dfrac{\mathbb{W}_1(\mu A_\mu, \nu A_\nu)}{\mathbb{W}_1(\mu, \nu)}$. Let $\mu \neq \nu \in \mathcal{P}(E)$, we have:

$$\frac{\mathbb{W}_1(\mu A_\mu, \nu A_\nu)}{\mathbb{W}_1(\mu, \nu)} \leq \frac{\mathbb{W}_1(\mu A_\mu, \nu A_\mu)}{\mathbb{W}_1(\mu, \nu)} + \frac{\mathbb{W}_1(\nu A_\mu, \nu A_\nu)}{\mathbb{W}_1(\mu, \nu)}$$

Let us start with the first term:

$$\frac{\mathbb{W}_1(\mu A_\mu, \nu A_\mu)}{\mathbb{W}_1(\mu, \nu)} \leq \frac{\mathbb{W}_1(\mu \Pi[\Psi_{G(\bullet, \bullet)}(\mu)L], \nu \Pi[\Psi_{G(\bullet, \bullet)}(\mu)L])}{\mathbb{W}_1(\mu, \nu)}$$

$$\leq \sup_{x \neq y} \frac{\mathbb{W}_1(\Pi[\Psi_{G(x, \bullet)}(\mu)L], \Pi[\Psi_{G(y, \bullet)}(\mu)L])}{d(x, y)}$$

$$\leq \tau_1(\Pi)\tau_1(L) \sup_{x \neq y} \frac{\mathbb{W}_1(\Psi_{G(x, \bullet)}(\mu), \Psi_{G(y, \bullet)}(\mu))}{d(x, y)}$$

$$\leq \tau_1(\Pi)\tau_1(L) \frac{2\|G\|_{Lip,\infty}\mathrm{diam}(E)}{\epsilon(G)},$$

where we used Lemma 1 for the second and third lines, and Propositions 5, 6 and 7 for the third and last. As for the second term, we have:

$$\mathbb{W}_1(\nu A_\mu, \nu A_\nu) = \mathbb{W}_1(\nu \Pi[\Psi_G(\mu)L], \nu \Pi[\Psi_G(\nu)L])$$

$$\leq \int \nu(dx)\mathbb{W}_1(\Pi[\Psi_{G(x, \bullet)}(\mu)L], \Pi[\Psi_{G(x, \bullet)}(\nu)L])$$

$$\leq \tau_1(\Pi)\tau_1(L) \int \nu(dx)\mathbb{W}_1(\Psi_{G(x, \bullet)}(\mu), \Psi_{G(x, \bullet)}(\nu))$$

$$\leq \tau_1(\Pi)\tau_1(L) \int \nu(dx) \frac{2\|G(x, \bullet)\|_{Lip}\mathrm{diam}(E)}{\epsilon(G)}\mathbb{W}_1(\mu, \nu)$$

$$\leq \tau_1(\Pi)\tau_1(L) \frac{2\|G\|_{\infty,Lip}\mathrm{diam}(E)}{\epsilon(G)}\mathbb{W}_1(\mu, \nu)$$

where we also used Lemma 1 for the second and third lines, and Propositions 5, 6 and 7 for the third and last. □

## C    Proofs From Section 5.2

**Lemma 2.** *For any $f : \mathbb{R}^d \to \mathbb{R}$, we have*

$$\|f\|_{Lip} = \sup_{x \neq y, \|x - y\| \leq 1} \frac{|f(x) - f(y)|}{\|x - y\|}. \tag{6}$$

*Proof.* Let $x \neq y$ and $L := \sup_{x \neq y, \|x - y\| \leq 1} \frac{|f(x) - f(y)|}{\|x - y\|} \leq \infty$. First, assume $\|f\|_{Lip}, L < \infty$. It is clear that $L \leq \|f\|_{Lip}$ since $\{x \neq y, \|x - y\| \leq 1\} \subset \{x \neq y\}$. For the reverse inequality, we split the segment $[x, y]$ into

the minimum number of chunks of lengths smaller than 1: $x = z_1 \to z_2 \to \cdots \to z_k = y$ (in particular, if $\|x - y\| \leq 1$ then $z_2 = y$). Then

$$|f(x) - f(y)| \leq \sum_{1 \leq i \leq k-1} |f(z_i) - f(z_{i+1})|$$

$$\leq L \sum_{1 \leq i \leq k-1} \|z_i - z_{i+1}\| = L\|x - y\|.$$

which gives $\|f\|_{Lip} \leq L$ so $L = \|f\|_{Lip}$. Now if $\|f\|_{Lip} = \infty$ but $L < \infty$, by applying the above argument we can obtain a contradiction. Finally, it suffices to note that the case where $\|f\|_{Lip} < \infty$ but $L = \infty$ is impossible since $\|f\|_{Lip} \geq L$. □

**Lemma 3.** *For any $n$ and $(z_1, \cdots, z_n) \in \mathbb{R}_+^n$:*

$$f(z_1, \cdots, z_n) := \frac{\sum_{i=1}^n z_i e^{-z_i^2}}{1 + \sum_{i=1}^n e^{-z_i^2}} \leq \sqrt{\ln n + \frac{1}{2e}}. \tag{7}$$

*Proof.* $f$ is clearly bounded on $\mathbb{R}_+^n$ ($z_i e^{-z_i^2} \to 0$ when $z_i \to \infty$). Let us now compute the partial derivatives of $f$. For a given $z_i$:

$$\frac{\partial f}{\partial z_i} = \frac{e^{-z_i^2}}{1 + \sum_{k=1}^n e^{-z_k^2}} [1 - 2z_i^2 + 2z_i f(z_1, \cdots, z_n)].$$

There is only one positive solution of $1 - 2z_i^2 + 2z_i f^* = 0$, meaning that $f$ reaches its maximum when all its coordinates are equal. We thus only need to study:

$$g(x) := \frac{nxe^{-x^2}}{1 + ne^{-x^2}} = \frac{xe^{\ln n - x^2}}{1 + e^{\ln n - x^2}}. \tag{8}$$

The change of variable $y = \ln n - x^2$ gives $g(y) = \frac{\sqrt{\ln n - y}e^y}{1 + e^y} \leq \frac{\sqrt{\ln n - y}}{1 + e^{-y}}$ with $y \in ]-\infty, \ln n]$.

On $[0, \ln n]$, we clearly have $g(y) \leq \sqrt{\ln n}$. Let us consider $y \in ]-\infty, 0]$. We get $g^2(y) = \frac{\ln n - y}{(1 + e^{-y})^2} \leq \frac{\ln n - y}{e^{-2y}} \leq \ln n + \frac{1}{2e}$ with since $(2e)^{-1}$ is the maximum of of $ze^{-2z}$ on $\mathbb{R}_+$. This concludes the proof. □

**Lemma 4.** *Let $\mu_1, \mu_2, \nu_1, \nu_2 \in \mathcal{W}_1(\mathbb{R}^d)$. Then*

$$\mathbb{W}_1(\mu_1 \otimes \mu_2, \nu_1 \otimes \nu_2) \leq \mathbb{W}_1(\mu_1, \nu_1) + \mathbb{W}_1(\mu_2, \nu_2)$$

*Proof.* Let $\gamma_1 \in \mathcal{C}(\mu_1, \nu_1), \gamma_2 \in \mathcal{C}(\mu_2, \nu_2)$ be optimal for $c(x, y) = \|x - y\|_1$. Note that $\gamma_1 \otimes \gamma_2 \in \mathcal{C}(\mu_1 \otimes \mu_2, \nu_1 \otimes \nu_2)$, i.e. $\gamma_1 \otimes \gamma_2$ is a transfer plan with the correct marginals, by considering

$$\int_{\mathbb{R}^d \times \mathbb{R}^d} \mathrm{d}\gamma_1 \otimes \gamma_2(x_1, x_2, y_1, y_y) = \int_{\mathbb{R}^d \times \mathbb{R}^d} \mathrm{d}\gamma_1(x_1, y_1)\mathrm{d}\gamma(x_2, y_2)$$

$$= \int_{\mathbb{R}^d} \mathrm{d}\gamma_1(x_1, y_1) \int_{\mathbb{R}^d} \mathrm{d}\gamma_2(x_2, y_2)$$

$$= \nu_1(\mathrm{d}y_1)\nu_2(\mathrm{d}y_2) = \mathrm{d}\nu_1 \otimes \nu_2(y_1, y_2)$$

and same for the other marginals.

Thus we have

$$
\begin{aligned}
\mathrm{W}_1(\mu_1 \otimes \mu_2, \nu_1 \otimes \nu_2) &= \inf_{\gamma \in \mathcal{C}(\mu_1 \otimes \mu_2, \nu_1 \otimes \nu_2)} \int \|(x_1, x_2) - (y_1, y_2)\| \mathrm{d}\gamma(x_1, x_2, y_1, y_2) \\
&= \inf_{\gamma \in \mathcal{C}(\mu_1 \otimes \mu_2, \nu_1 \otimes \nu_2)} \int (\|x_1 - y_1\| + \|y_1, y_2\|) \mathrm{d}\gamma(x_1, x_2, y_1, y_2) \\
&= \inf_{\gamma \in \mathcal{C}(\mu_1 \otimes \mu_2, \nu_1 \otimes \nu_2)} \int \|x_1 - y_1\| \mathrm{d}\gamma(x_1, x_2, y_1, y_2) + \cdots \\
&\quad \cdots + \inf_{\gamma \in \mathcal{C}(\mu_1 \otimes \mu_2, \nu_1 \otimes \nu_2)} \int \|x_2 - y_2\| \mathrm{d}\gamma(x_1, x_2, y_1, y_2) \\
&\leq \int \|x_1 - y_1\| \mathrm{d}\gamma_1 \otimes \gamma_2(x_1, x_2, y_1, y_2) + \int \|x_2 - y_2\| \mathrm{d}\gamma_1 \otimes \gamma_2(x_1, x_2, y_1, y_2) \\
&= \int \|x_1 - y_1\| \mathrm{d}\gamma_1(x_1, y_1) + \int \|x_2 - y_2\| \mathrm{d}\gamma_2(x_2, y_2) \\
&= \mathrm{W}_1(\mu_1, \nu_1) + \mathrm{W}_1(\mu_2, \nu_2)
\end{aligned}
$$

$\square$

**Proposition 8.** *Let $E = \mathbb{R}^d$ and suppose $X = \{x_1, \ldots, x_N\}$ and $Y = \{y_1, \ldots, y_N\}$ Let $\mu = m(X)$, $\nu = m(Y)$. Then for $x \in supp(\mu)$ and $y \in supp(\nu)$, we have*

$$
\mathrm{W}_1(\Psi_{G(x, \bullet)}(\mu), \Psi_{G(y, \bullet)}(\nu)) \leq \left[ \sqrt{d} \sqrt{\ln N + \frac{1}{2e}} \|G\|_{Lip} + \|G\|_\infty + \sqrt{d} + 2 \right] (d(x, y) + \mathrm{W}_1(\mu, \nu)).
$$

*Proof.* We use the Kantorovich formulation of $\mathrm{W}_1$. Let $f$ be a function with $\|f\|_{Lip} \leq 1$. Using the same kind of technique as in Section B, we can assume without loss of generality that $f(y) = 0$. For simplicity, we write $G(x, \bullet) = G_x$. We wish to upper-bound the quantity $|\Psi_{G_x}(\mu)(f) - \Psi_{G_y}(\nu)(f)|$.

Because $\Psi_{G_x}$ and $\Psi_{G_y}$ are homonegeous in their measure argument, and for the sake of simplicity, we write $\mu = \sum_i \delta_{x_i}$ $\nu = \sum_i \delta_{y_i}$ (which is equivalent to simplifying by $1/N$ in e.g. the numerator and denominator of $\Psi_{G_x}$). This guarantees in particular that $\mu(G_x) \geq 1$ and $\nu(G_y) \geq 1$ ($x$ and $y$ are in $supp(\mu)$ and $supp(\nu)$ resp.) and equivalently that $1/\mu(G_x) \leq 1$ and $1/\nu(G_y) \leq 1$.

Then:

$$
\begin{aligned}
|\Psi_{G_x}(\mu)(f) - \Psi_{G_y}(\nu)(f)| &= \frac{1}{\mu(G_x)\nu(G_y)} |\nu(G_y)\mu(G_x f) - \mu(G_x)\nu(G_y f)| \\
&= \frac{1}{\mu(G_x)\nu(G_y)} |\nu(G_y)\mu(G_x f) - \nu(G_y)\nu(G_y f) + \nu(G_y)\nu(G_y f) - \mu(G_x)\nu(G_y f)| \\
&\leq \frac{\nu(G_y)}{\mu(G_x)\nu(G_y)} |\mu(G_x f) - \nu(G_y f)| + \frac{\nu(G_y f)}{\mu(G_x)\nu(G_y)} |\nu(G_y) - \mu(G_x)|. \quad (9)
\end{aligned}
$$

We start by bounding the second term of (9). We have:

$$
\begin{aligned}
\frac{\nu(G_y f)}{\mu(G_x)\nu(G_y)} |\nu(G_y) - \mu(G_x)| &= \frac{\nu(G_y f)}{\mu(G_x)\nu(G_y)} |(\delta_x \otimes \mu)(G) - (\delta_y \otimes \nu)(G)| \\
&\leq \frac{\nu(G_y f)}{\mu(G_x)\nu(G_y)} \|G\|_{Lip} \mathrm{W}_1(\delta_x \otimes \mu, \delta_y \otimes \nu).
\end{aligned}
$$

Here, $\delta_x \otimes \mu$ denotes the product of the two measures on $E \times E$. Since $f(y) = 0$, we see that $f(z) \le f(y) + \|f\|_{Lip}\|y - z\|_1 \le \|y - z\|_1$. This gives:

$$\frac{\nu(G_y f)}{\nu(G_y)} = \frac{\int G_y(z)f(z)\nu(\mathrm{d}z)}{\int G_y(z)\nu(\mathrm{d}z)} \le \frac{\int G_y(z)\|y-z\|_1 \nu(\mathrm{d}z)}{\int G_y(z)\nu(\mathrm{d}z)}$$

$$\le \frac{\sum_{i=1}^{N} G(y, y_i)\|y - y_i\|_1}{\sum_{i=1}^{N} G(y, y_i)} \le \sqrt{d}\frac{\sum_{i=1}^{N} e^{-\|y-y_i\|_2^2}\|y - y_i\|_2}{\sum_{i=1}^{N} e^{-\|y-y_i\|_2^2}},$$

where we applied Cauchy-Schwartz for the last inequality. Since $y = y_i$ for a given $i$, we are interested in the quantity $\frac{\sum_{i=1}^{N-1} z_i e^{-z_i^2}}{1 + \sum_{i=1}^{N-1} e^{-z_i^2}}$ for arbitrary $z_i \ge 0$. Applying Lemma 3 with $n = N - 1$ gives an upper-bound of $\sqrt{\ln N + \frac{1}{2e}}$.

Let us now consider the first term of (9):

$$\frac{\nu(G_y)}{\mu(G_x)\nu(G_y)}|\mu(G_x f) - \nu(G_y f)| = \frac{1}{\mu(G_x)}|\mu(G_x f) - \nu(G_y f)|$$

$$\le \frac{1}{\mu(G_x)}\|Gf\|_{Lip}\mathrm{W}_1(\delta_x \otimes \mu, \delta_y \otimes \nu).$$

To estimate $\|Gf\|_{Lip}$ we have

$$\|Gf\|_{Lip} = \sup_{(x,w) \ne (y,z)} \frac{|G(x,w)f(w) - G(y,z)f(z)|}{\|(x,w) - (y,z)\|_1}$$

where additionally, we can assume that $\|(x,w) - (y,z)\| \le 1$ (see Lemma 2). We have:

$$|G(x,w)f(w) - G(y,z)f(z)| = |G(x,w)f(w) - G(x,w)f(z) + G(x,w)f(z) - G(y,z)f(z)|$$
$$\le |G(x,w)||f(w) - f(z)| + |f(z)||G(x,w) - G(y,z)|.$$

For the first term, we see that

$$|G(x,w)||f(w) - f(z)| \le \|G\|_{\infty,\infty}\|f\|_{Lip}d(w,z)$$
$$\le \|G\|_{\infty,\infty}\|f\|_{Lip}(d(w,z) + d(x,y)).$$

For the second term, we have

$$|f(z)||G(x,w) - G(y,z)| \le \|y - z\|_1|G(x,w) - G(y,z)|$$
$$\le \|y - z\|_1\|\nabla G(t_1, t_2))\|_\infty\|(x,w) - (y,z)\|_1,$$

for $t_1$ in the segment $[x, y]$ and $t_2$ in the segment $[w, z]$ (this follows directly from the mean value theorem, note that the gradient is taken with respect to both variables). We used $f(y) = 0$ and $f(z) \le f(y) + \|f\|_{Lip}\|y - z\|_1 = \|y - z\|_1$ in the first line.

In the Gaussian case:

$$\|y - z\|_1\|\nabla G(t_1, t_2))\|_\infty \le (\|y - t_1\|_1 + \|t_1 - t_2\|_1 + \|t_2 - z\|_1)2\|t_1 - t_2\|_\infty e^{-\|t_1-t_2\|_2^2}$$

$$\le 2(2 + \|t_1 - t_2\|_1)\|t_1 - t_2\|_\infty e^{-\|t_1-t_2\|_2^2},$$

where we used the fact that $\|y - t_1\|_1 \le 1$ and $\|t_2 - z\|_1 \le 1$ ($t_1$ is in the $[x, y]$ segment and $\|x - y\|_1 \le 1$ by assumption). That upper bound is uniformly bounded with respect to $t_1$ and $t_2$, we let $C$ denote that constant. A loose upper-bound on $C$ is $\sqrt{d} + 2$ (which we use in the statement of the proposition).

To conclude, it suffices to note that by Lemma 4 we have

$$\mathrm{W}_1(\delta_x \otimes \mu, \delta_y \otimes \nu) \le \mathrm{W}_1(\delta_x, \delta_y) + \mathrm{W}_1(\mu, \nu).$$

$\square$

**Theorem 2.** *Let $E=\mathbb{R}^d$ and suppose $X = \{x_1, \ldots, x_N\}, Y = \{y_1, \ldots, y_M\} \subset \mathbb{R}^d$. Let $G(x,y) = \exp(-\|x - y\|_2^2)$ and $\Pi$ be the usual projection onto $\mathcal{P}_\delta(\mathbb{R}^d)$. Then for $\mu = m(X)$ and $\nu = m(Y)$,*

$$\mathbb{W}_1(\mu \mathbf{A}_\mu, \nu \mathbf{A}_\nu) \leq 2\tau(\Pi)\tau(L)\left[\|G\|_\infty + \sqrt{d} + 2 + \sqrt{d}\sqrt{\ln(\min(N,M)) + \frac{1}{2e}}\|G\|_{Lip}\right]\mathbb{W}_1(\mu, \nu).$$

*Proof.* Firstly, using Proposition 1, we know that $\mu \mathbf{A}_\mu$ is another empirical measure concentrated on $\{\text{Attention}(x_i, X, X)\}$, similarly, $\nu \mathbf{A}_\nu$ is concentrated on $\{\text{Attention}(y_i, Y, Y)\}$. This fact allows us to use the following result from Santambrogio (2015) Equation 6.2

$$\mathbb{W}_1(\mu, \nu) = \min\left\{\sum_{i,j} \gamma_{ij} d(x_i, y_j) \mid \gamma_{i,j} \geq 0, \ \sum_i \gamma_{ij} = \frac{1}{M}, \ \sum_j \gamma_{ij} = \frac{1}{N}\right\},$$

Applied to $\mathbb{W}_1(\mu \mathbf{A}_\mu, \nu \mathbf{A}_\nu)$, it gives

$$\mathbb{W}_1(\mu \mathbf{A}_\mu, \nu \mathbf{A}_\nu) = \min\left\{\sum_{i,j} \gamma_{ij} d(\text{Attention}(x_i, X, X), \text{Attention}(y_j, Y, Y)) \mid \right.$$
$$\left. \gamma_{i,j} \geq 0, \ \sum_i \gamma_{ij} = \frac{1}{M}, \ \sum_j \gamma_{ij} = \frac{1}{N}\right\}$$
$$= \min\left\{\sum_{i,j} \gamma_{ij} \mathbb{W}_1(\mathbf{A}_\mu(x_i, \bullet), \mathbf{A}_\nu(y_i, \bullet)) \mid \right.$$
$$\left. \gamma_{i,j} \geq 0, \ \sum_i \gamma_{ij} = \frac{1}{M}, \ \sum_j \gamma_{ij} = \frac{1}{N}\right\}.$$

Using Lemma 1 for each term, we have

$$\mathbb{W}_1(\mathbf{A}_\mu(x_i, \bullet), \mathbf{A}_\nu(y_j, \bullet)) \leq \tau(\Pi)\tau(L)\mathbb{W}_1(\Psi_{G(x_i, \bullet)}(\mu), \Psi_{G(y_j, \bullet)}(\nu)).$$

Now, from Proposition 8 ($x_i$ belongs to supp $(\mu)$ and $y_j$ to supp $(\nu)$), we get

$$\mathbb{W}_1(\Psi_{G(x_i, \bullet)}(\mu), \Psi_{G(y_j, \bullet)}(\nu))$$
$$\leq \left[\sqrt{d}\sqrt{\ln N + \frac{1}{2e}}\|G\|_{Lip} + \|G\|_\infty + \sqrt{d} + 2\right](d(x_i, y_j) + \mathbb{W}_1(\mu, \nu)).$$

Substituting this back into the above formula, we obtain

$$
\mathbb{W}_1(\mu \mathbf{A}_\mu, \nu \mathbf{A}_\nu)
$$
$$
\leq \min \Big\{ \sum_{i,j} \gamma_{ij} \mathbb{W}_1(\mathbf{A}_\mu(x_i, \bullet), \mathbf{A}_\nu(y_i, \bullet)) \mid \gamma_{i,j} \geq 0, \ \sum_i \gamma_{ij} = \frac{1}{M}, \ \sum_j \gamma_{ij} = \frac{1}{N} \Big\}
$$
$$
\leq \tau(\Pi)\tau(L) \min \Big\{ \sum_{i,j} \gamma_{ij} \Big[ \sqrt{d}\sqrt{\ln N + \frac{1}{2e}} \|G\|_{Lip} + \|G\|_\infty + \sqrt{d} + 2 \Big] (d(x_i, y_j) + \mathbb{W}_1(\mu, \nu)) \mid
$$
$$
\gamma_{i,j} \geq 0, \ \sum_i \gamma_{ij} = \frac{1}{M}, \ \sum_j \gamma_{ij} = \frac{1}{N} \Big\}
$$
$$
= \tau(\Pi)\tau(L) \Big[ \sqrt{d}\sqrt{\ln N + \frac{1}{2e}} \|G\|_{Lip} + \|G\|_\infty + \sqrt{d} + 2 \Big] \Big( \mathbb{W}_1(\mu, \nu) +
$$
$$
\min \Big\{ \sum_{i,j} \gamma_{ij} d(x_i, y_j) \mid \gamma_{i,j} \geq 0, \ \sum_i \gamma_{ij} = \frac{1}{M}, \ \sum_j \gamma_{ij} = \frac{1}{N} \Big\} \Big)
$$
$$
= \tau(\Pi)\tau(L) \Big[ \sqrt{d}\sqrt{\ln N + \frac{1}{2e}} \|G\|_{Lip} + \|G\|_\infty + \sqrt{d} + 2 \Big] (\mathbb{W}_1(\mu, \nu) + \mathbb{W}_1(\mu, \nu))
$$
$$
= 2\tau(\Pi)\tau(L) \Big[ \sqrt{d}\sqrt{\ln N + \frac{1}{2e}} \|G\|_{Lip} + \|G\|_\infty + \sqrt{d} + 2 \Big] \mathbb{W}_1(\mu, \nu),
$$

where we used in particular $\sum_{i,j} \gamma_{ij} = 1$. The inequality being valid for both $M$ and $N$, taking the min gives the result.

$\square$

## D  Preliminary Experiments

Below is an analysis for the "regularity mismatch" hypothesis from Section 6.2 on the modified "ConceptNet" dataset from Tenney et al. (2019). We measure the 1-Wasserstein distance between original and negated input sentences and relate that distance to the amount of overlap between the model's prediction on those sentences. We use a pretrained BERT-base model Devlin et al. (2018) without modification to the model. This model includes confounding factors such as LayerNorm, residual connections, and feed-forward networks, but we feel it can be illustrative nonetheless.

Firstly, we see that using the 1-Wasserstein distance to compare inputs of different lengths is sensible; it reveals that ConceptNet has two distinct negation types (see Figure 2(b)) producing different input perturbation distances (x-axes in Figure 2(a)). Secondly, the regularity of self-attention networks is clearly demonstrated by the positive correlation between input distance and output distance in Figure 2(a). Finally, we find that the mean perturbation distance is statistically significantly larger ($\alpha = 0.05$) when the model makes a different prediction under negation than when the model makes the same prediction (Fig. 2(c)).

This demonstrates that our theory can be used to design sensible experiments and make predictions about what to expect from them. Much more work in this direction is necessary to fully explore the potential of this paradigm.

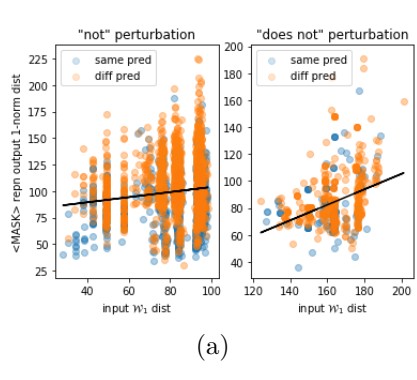

(a)

| negation type | original | perturbed |
|---|---|---|
| "not" | A banjo is made of [MASK]. | A banjo is **not** made of [MASK]. |
| "does not" | An airplane requires [MASK]. | An airplane **does not require** [MASK]. |

(b)

| negation | pred | N | % same | mean $\mathbb{W}_1$ | $p$ value |
|---|---|---|---|---|---|
| "not" | same | 2206 | 32% | 80.31 | 1.09E-11 |
| | diff | 4650 | | 83.28 | (reject $H_0$) |
| "does not" | same | 402 | 28% | 162.16 | 0.0346 |
| | diff | 1038 | | 163.32 | (reject $H_0$) |

(c)

**Figure 2:** Results for "negation" experiment proposed in Section 5.3 of our paper using the ConceptNet dataset from Tenney et al. (2019). (a) 1-norm distance of `[MASK]` token representations in (used for prediction) vs input sentence $\mathbb{W}_1$ distance. (b) typical examples of both types of negation perturbation. (c) experimental results collected; $\alpha = 0.05$.

