# OpenReview forum: "On the Regularity of Attention"
_TMLR — Rejected by TMLR_

### Review · Reviewer_q9uW · 2022-11-25

**Summary Of Contributions:**

The paper proposes a new framework in which the attention mechanism is defined as an operator acting on empirical measures over representations of tokens. Further, the space of representations is equipped with the 1-Wasserstein distance that allows assessing the regularity of attention in terms of its Lipschitz continuity. As a result, a theoretical analysis of attention is presented.

**Audience:**

No

**Broader Impact Concerns:**

None.

**Claims And Evidence:**

No

**Requested Changes:**

- Stating the problem and claims clearly would greatly help to read the paper.
- The structure of the paper could be improved.
- More intuition should be provided.
- Remark 2: Please correct the citation.

**Strengths And Weaknesses:**

**Strengths**

- The provided theoretical analysis may be interesting.

**Weaknesses**

- The paper is of a theoretical character, thus, it is very formal. However, sometimes it is hard to follow because very little intuition is provided. Basically, the paper consists mostly of definitions, remarks, and propositions. It is not accessible to an average ML researcher. As a result, the flow of the paper is quite difficult to follow. The paper seems like a combination of various ideas (e.g., rephrasing attention in the measure-theoretic language, the regularity of attention) that makes the paper even harder to read.
- Remark 1 (namely, the most general case of attention) is not necessarily straightforward. Would it be possible to provide a better intuition why the provided \ell function is indeed the most general case?
- Applications presented in Section 6 are quite unclear to me. First, as stated by the authors: “our framework provides a potential mathematical basis for this phenomenon [robustness to noisy inputs and adversarial examples] in transformers”. Since it’s a potential explanation, it is not convincing. Second, a similar statement (“our modelling could potentially be used (…)” about negated sentences again is not necessarily interesting since it is only a potential explanation.

---

> ### Author Response · Authors · 2023-03-11
> **Thank you for your feedback!**
>
> Please refer to our general comments above regarding readability.
>
> **The paper seems like a combination of various ideas (e.g., rephrasing attention in the measure-theoretic language, the regularity of attention)**. The main objective of the paper is to study the regularity of attention; our approach relies on rephrasing attention in the measure-theoretic language. The (single) main  idea of the paper is to exploit a  mapping of the inputs of attention to empirical distributions and to derive Lipschitz bounds using the  Wasserstein distance.
>
> **On Remark 1**. The expression we give is merely a (admittedly slighlty pedantic) way to encode *any* correspondence v_i <--> k_i : it just says l(k) = 0 ig k is not one of the k_i and l(k_i) = v_i.  Most most realistic implementations use explicit modelling choices,  such as a linear correspondence l(k) = W k.
>
> **Section 6**. We agree this section lacks the rigour of the previous ones. It is in fact meant to be a discussion / outlook / motivation section. We changed its title to clarify this.

---

### Review · Reviewer_H2QJ · 2022-12-08

**Summary Of Contributions:**

The paper introduces a new mathematical framework from measure theory and integral operators to study the regularity of attention, the paper focuses on the detailed layout of the theoretical framework and later leverage the constructed of the framework to show that attention is Lipschitz (under assumptions), and offers other additional discussions. Overall, I think this paper has great potential in studying the attention mechanism, but seems further clarification will be beneficial at this moment.

**Audience:**

Yes

**Broader Impact Concerns:**

none noted.

**Claims And Evidence:**

Yes

**Requested Changes:**

As a summary of above, please try to (if possible)

 - provide more empirical evidence to show that the theoretical analysis are meaningful, for example, do we have practical evidence that the attention is Lipschitz?
 - offer more rigorous discussions in Section 6
 - offer more intuitive discussions of what the empirical/practical implications of the assumptions used are.

**Strengths And Weaknesses:**

- strengths
    - this paper, in my opinion, has great potential in introducing a new perspective of studying the attention mechanism
    - the mathematical discussion looks rigorous and readable, with assumptions clearly discussed in rigorous terms.
- weakness
    - the most important question, in my opinion, is that attention is practice is probably not Lipschitz, in fact, the existence of adversarial examples probably denies that the function is Lipschitz, or maybe the authors believe the adversarial examples are introduced mostly in other components in the model?
       - some more convincing empirical results on this regard is probably needed.
    - while this paper aims to study the attention in deep learning, particularly transformers, the linkage to the actual attention module seems very weak at this moment.
       - it seems to me the primary linkage to the empirical discussion of transformers happens in Section 6, where most of the connections are discussions on what can be explained if continued proofs are performed.
   - it is probably necessary to intuitively explain the assumptions needed (what empirical implications these assumptions have)
       - without such discussions, it's hard to evaluate the actual application value of the theory.

---

> ### Author Response · Authors · 2023-03-11
> **Thank you for your feedback!**
>
> **On Lipschitzness and adversarial examples**. One motivation for understanding and controlling the Lipschitzness properties of neural nertworks (e.g. through controllable upper bounds)  is indeed to improve improve robustness against adversarial perturbation (e.g. by imposing explicit Lipschitzness constraints). The goal of the paper is to carry out such a theoretical study in the context of attention.
>
> **it seems to me the primary linkage to the empirical discussion of transformers happens in Section 6, where most of the connections are discussions on what can be explained if continued proofs are performed**. That is true --  but that is also not the purpose of the paper. We have changed the title of Section 6 (now "Discussion")  to clarify its status / role in the paper.

---

### Review · Reviewer_fWGp · 2023-02-08

**Summary Of Contributions:**

This paper proposes a measure-theoretic generalization of attention and analyzes the regularity properties of this "operator" attention.

**Audience:**

Yes

**Claims And Evidence:**

Yes

**Requested Changes:**

In addition to the above weaknesses, which I want the authors to address, I have several other miscellaneous concerns. Broadly, I hope the authors can edit the manuscript to correct for typos and to strengthen writing. However, some important comments include:

* "Softmatch" is incorrect, "softmax" is the commonly used term.
* The current formulation of operator acting on a measure $\mu A$ is unclear; rewrite to make it clear that $\mu$ is being acted on and that this produces a new measure afterwards. Generally, the authors should do this for several other core kernel operations.
* Generally, I would suggest the author check their syntax. In particular, it is very difficult to follow a lot of the math due to an overload of symbols. For example, the equation at the top of page 6 highlights how unnecessarily cluttered the writing is. Removing symbols and defining other operations would drastically cut this down.
* Several citations are misconstructed. For example, Smola and Zhang does not have enough details, and Lu et al. needs a date.
* Figure 1 is entirely unparseable and should be simplified.
* Section 6.2 needs to be made more rigorous.

**Strengths And Weaknesses:**

Strengths
-----------
* The paper is technically correct.

Weaknesses
---------------
* The overriding criticism I have with this work is that the introduced generalization to measure space feels entirely unnecessary and seems to be done for purely aesthetic reasons. In particular, no one works with a measure-theoretic notion of attention, so introducing it here for analysis seems entirely superfluous unless some greater insight can be derived. Notably, the Lipschitzness analyses are direct generalizations of those done by Kim et al. 2021 to the measure theoretic case, although a modicum of care must be taken to do this. As a result, the applications heavily mirror those from Kim et al. 2021 and, in particular, do not seem to need the measure-theoretic framework. I would suggest there be a more direct section justifying and explicating the use cases of the measure-theoretic framework, rather than the current vague justifications that are sprinkled throughout the paper.

---

> ### Author Response · Authors · 2023-03-11
> **Thanks for your feedback!**
>
> Please refer to our general comments above regarding novelty and readability
>
> **"Softmatch vs softmax.** Softmax is an operation on a vector, softmatch is an operation on a query and some keys (note that in Eq 1, softmatch(q, K)  := softmax(a(q, K))).
>
> **The current formulation of operator acting on a measure μA is unclear** This is a standard notation for Markov kernels though (e.g., del Moral 2004). Besides changing notation (to what?), we're not so sure how we can be clearer than making the definition of  the operation P(E) --> P(E) explicit as we do in Section 3.2.
>
> **Removing symbols and defining other operations would drastically cut this down** Do you have a concrete example of alternative definitions and symbols that would improve the flow?
>
> **misconstructed citations** Thank you for catching this (note that Smola and Zhang is a slide deck though).
>
> **Figure 1 is entirely unparseable and should be simplified** Concrete suggestions would help us here. We did our best to make Fig 1 as parseable as possible...

---

### Review · Reviewer_fXy8 · 2023-02-09

**Summary Of Contributions:**

This paper presents a new mathematical framework that models the attention mechanism for sequence modelling in neural networks. The framework treats sequences as empirical measures over the representations of tokens in the sequence, and the mathematical tools used for this are introduced in sections 3 and 4.

Using this new framework, in section 5 the authors demonstrate that the attention mechanism is regular (i.e. lipschitz continuous) with respect to the 1-Wasserstein distance. The authors first show this in the case of bounded representation space for dot-product attention (Theorem 1), before extending to unbounded representation space using a different attention (so-called 'Gaussian interaction potential' similarity instead of dot-product. These results closely mirror and are inspired by existing work of Kim et al 2021. Besides offering a different proof mechanism, the authors claim that their framework has the additional advantage of being agnostic to the length of the sequence.

The authors then argue briefly in section 6 that their analysis on the regularity of attention can be applied to concrete problems such as robustness and invertible/infinite-depth transformers.

**Audience:**

Yes

**Claims And Evidence:**

Yes

**Requested Changes:**

Please correct all the typos to improve readability, and respond to my weaknesses/questions above.

**Strengths And Weaknesses:**

Strengths:
1. The mathematical framework of considering sequences as empirical measures, and considering neural network (NN) layers, such as attention layers, through their actions on such empirical measures, is novel as far as I am aware, and could be useful in improving our understanding of NNs that model sequences. This is potentially pertinent given the popularity of transformers.
2. The question of regularity in neural networks, in particular ones that use attention, is a topic of interest to some in the community. This is also a weakness as many of the results concerning regularity have appeared in similar flavour in prior work (e.g. Kim et al, 2021).
3. The ability to model distances between sequences of various lengths is the main strength of the authors' framework, compared to previous methods that use Jacobian based arguments to consider regularity (e.g. Kim et al, 2021).

Weaknesses:
1. The novelty of the results in the paper seems limited, as discussed above. It is not clear the benefit (e.g. useful insight) that one gains from being able to compare sequences of different lengths.
2. Likewise, it is not clear that comparing sequences through abstractions to some empirical measure space is a meaningful or natural way of doing so. The authors state themselves that: 'Section 5 relies on the 1-Wasserstein distance, which is trivial in the discrete case' (compared to this measure space), is it not problematic if the results depend on the choice of abstraction through which we view the sequence? Moreover, is it even the case that the mapping between sequence <-> empirical measure is one-to-one? You could have sequence 1: 'me', and sequence 2: 'meme', and for a tokeniser which splits seq1 to be length one, ['me'], and seq2 of length two, ['me', 'me'], which would give the same empirical measures for these different words right (both are just a single dirac delta at 'me')?
3. I did not find the discussions in section 6 about the 'applications of regularity' to be particularly compelling. As the authors point out, much of these points (e.g. robustness or invertibility) are raised in previous work e.g. Kim et al. 2021 and it is not clear what new perspective is being offered in this work. Also, these applications could do with some more thorough analysis or experimental evidence if the authors wish to strengthen the contribution of section 6. Finally, I would move 6.1 into section 5: Prop 3 doesn't really appear to be an application per se but rather the Key-Value equivalent for Corr 1 in section 5.
4. I imagine that the discussion of measure theory and integral operators (which feature prominently in sections 3 and 4) to be intimidating to some readers, which will likely make the work less accessible/impactful. Some cases this may be unavoidable, but in other cases this may hinder the readability of the work e.g. the introduction of Boltzmann-Gibbs transformations is really just a reweighted density if I am not mistaken?
5. I found some parts of the paper poorly presented (see below), which detracted from my reading/ability to understand the paper.

Typos/poor presentation:
1. Figure 1 (bottom left): shouldn't the 4th cylinder be red, not the 5th.
2. Figure 1 caption: why is $n$ used as the dimension of the representation vectors used, whereas it is $d$ in the rest of the paper?
3. Figure 1 caption: it would be better for the reader to introduce what $\tau$ is in the caption itself, rather than refer to section 6. Space is not an issue here
4. Bottom of page 2: 'rank collapse' is introduced without any explanation or context.
5. Top of page 3: Lu et al. citation should have a year?
6. Definition 2: 'such that and' -> 'such that'
7. Bottom of page 3: 'Markov transport' introduced without explanation.
8. Top of page 4: 'Botzmann' -> 'Boltzmann'
9. There seems to be some inconsistency betwee definitions of $L$ and $\Pi$. In page 5 $L$ is a way to transport between keys to values, and $\Pi$ is a way to average/project over values, whereas at the top of page 7 $L$ is now treated as the identity (or mapping key index to value index), and $\Pi$ seems to do both the averaging as well as obtaining the values? Also, the $V'$ notation is introduced and not used which is confusing
10. Section 6.1: 'This is used in in' has repeated 'in'.
11. Section 6.1: Proposition 5 is introduced without reference to appendix. (likewise proposition 2 is not proven, but i believe is a subcase of proposition 4, this should be made clear).
12. Page 12: 'we can apply our our' repeated 'our'

General questions/thoughts:
1. What is the motivation for the 1-Wasserstein distance, as opposed to p-Wasserstein distances or other distances over measures?
2. How can you obtain dependence on the particular values of K, V in say Corollary 1. For example, if all the values V are the same then the lipschitz constant should be 0 right? This dependence seems important for us to be able to use this type of analysis to improve our understanding of transformers/self-attention.
3. Is it clear that $|| \ell||_{Lip}$ (in Corollary 1) will be small/controllable in general in non self-attention cases? What about when we use projection matrices to map our keys and values to as is standard?

---

> ### Author Response · Authors · 2023-03-11
> **Thank you for your thorough feedback.**
>
> Please refer to our general comments above regarding novelty and readability.
>
> **You could have sequence 1: 'me', and sequence 2: 'meme', and for a tokeniser which splits seq1 to be length one, ['me'], and seq2 of length two, ['me', 'me']**
>
> This is a very good point,  we agree with you here. We are not sure that our framework can handle these these pathological situations. This is indeed a limitation of our framework.
>
> **Some cases this may be unavoidable, but in other cases this may hinder the readability of the work e.g. the introduction of Boltzmann-Gibbs transformations is really just a reweighted density if I am not mistaken?**
>
> Beside the fact that Diracs do not have densities, the gain in trading 'reweighted measures' for 'reweighted densities' is not clear to us. But more generally, we acknowledge that framing the whole framework in a language more familiar to the typical ML reader (assuming we would not loose too much in precision what we would gain in readability) could have been a useful contribution. This is not the editorial choice we've made: we found more natural to refer to measures and markov kernels using proper measure-theoretic  language (with Del Moral 2004 as our main textbook reference).  The question is whether this should form the basis for rejecting the paper for publication in TMRL  (we think this should not).
>
> **Typos / imprecisions**:  thanks for catching these. We have corrected most of them in the revised version.
>
> **inconsistency betwee definitions**. We disagree: p7 specializes the definition of p5 to the discrete case. The goal of this paragraph entitled "Recovering the Traditional Definition" is to explicitly show how to recover the traditional linear-algebraic definition of attention from our framework.
>
> **motivation for the 1-Wasserstein distance**: mainly the analytical tractability, due in particular  to the existence of the dual representation. Regarding the dependence of the results on the choice of abstraction through which we view the sequence: to us it is analogous to a dependence of regularity results on the choice of norm/metric. This choice is often dictated by analytical tractability and tightness of the obtained bounds.
>
> **Is it clear that \ell__lip (in Corollary 1) will be small/controllable in general in non self-attention cases?** this is a property of the function $\ell$, hence of a modelling choice. In the case of projection matrices, the Lipschitz constant is related to the matrix norm.

---

### Author Response · Authors · 2023-03-10
**General comments**

We thank all reviewers for their time and constructive feedback.

We would like to briefly comment on a couple of concerns shared by most reviewers: one is about novelty, in particular with respect to Kim et al 2021; another is readability, due to the formal nature of the paper.

**Regarding the comparison with Kim et al, 2021**.

We feel it might be worth pointing out that both works are actually **concurrent** (the first version of the two papers were made available on arXiv a few weeks apart). Since we are not quite sure about prior work policy for TMLR, we propose to give the non-anonymous details to the Action Editor,  in case this information can be useful. This should at least make clear that our results were not "inspired" by Kim et al, nor purposely designed to "closely mirror" or "directly generalize" their results. To our knowledge, both their work and ours were the first ones to investigate the regularity properties of attention.

 We also note that, according to TMLR guidelines https://jmlr.org/tmlr/acceptance-criteria.html, novelty and significance are not necessary criteria for acceptance. Now, interest to the community is one. We agree with reviewer fWGp that if our measure-theoretical approach was indeed "entirely unnecessary" and merely esthetical, the interest to the community would be limited. However, we emphasize that we obtain results that complement Kim et al's:  for example, while they obtain Lischitz bounds w.r.t the L2 and Linfinity norm, we obtain a Lipschitz bound w.r.t the L1 norm; our framework can handle inputs of arbitrary length; our framework can handle non gaussian potential functions. Finally, we believe the framing of self attention in a language that naturally describes interacting particle systems (Del Moral, 2004) is of interest in its own right -- this analogy was also drawn in the prior work (Lu et al, 2019) by other means.

**Regarding the readability of the paper**

We would agree with Reviewer q9uW that the formal nature of our measure-theoretical approach may make "the flow of the paper [...] quite difficult to follow" for the "average ML researcher". However, we feel this should not form the basis for rejecting the paper. Note that we are using the standard terminology of measure theory and markov kernels. It is not clear to us that trying to refer to measures and markov kernels  without using their bona fide definitions and the proper language for it would really improve readability.

**Summary of changes**

We thank all reviewers for catching misprints and imprecisions in the text, the figures and the citations.  We tried to incorporate most of the suggestions in the new version.

Section 6 was subject to criticisms due to its lack of precision / rigour compared to the rest of the paper. We simply chose to clarify its role in the paper, by changing the title from " Applications of Regularity" to "Discussion". We feel that dwelling further into the various points made in that section is beyond the scope of the paper. The goal in this section  is rather to illustrate the potential use and significance of regularity results in general and of our approach in particular.

---

### Decision · Action_Editors · 2023-04-05

**Recommendation:** Reject

**Comment:**

This paper introduces a measure-theoretic framework for studying the attention mechanism in neural networks. By formulating attention as an operator acting on empirical measures over representations of tokens, it is shown that attention is Lipschitz continuous with respect to the 1-Wasserstein distance.

While the reviewers appreciated many aspects of this paper, especially the rigor and clarity of the technical arguments, they were in agreement that it ultimately falls short of TMLR's acceptance criteria. First, some reviewers felt that the main claims were not supported by sufficient evidence, e.g. there was not convincing evidence to support the claim that proposed a methodology contributes a fundamental insight relevant for attention as it is used in practice.

Additionally, the reviewers questioned whether the paper would be of interest to parts of the TMLR community, citing a perhaps over-complicated formulation without clear benefits or useful takeaways. The authors responded to these criticisms in the discussion, noting several of the novel results made possible by their approach. I agree that there may be conclusions and insights from this work that could interest the community, but the current manuscript does not present them in a convincing way.

**Audience:**

While the audience of TMLR is broad and there may be individuals who find merit in the mathematical analysis performed in this paper, it is less clear whether there would be interest in the findings of this paper as they pertain to machine learning. In particular, the reviewers felt that the formulation was overly complicated without clearly articulated benefits, and they were not sure if there were any useful takeaways, either for theory or practice.

**Claims And Evidence:**

Some of the claims in this paper are indeed supported by convincing evidence, e.g. the reviewers agree that main theorems are presented clearly and proved rigorously. Some of the higher-level claims, however, are not be supported so convincingly. For example, it was at least implied that the proposed methodology contributes a fundamental insight relevant for attention as it is used in practice, and the reviewers did not find sufficient evidence to support this claim.